# Directional Privacy for Deep Learning

## Abstract

Differentially Private Stochastic Gradient Descent (DP-SGD) is a key method for applying privacy in the training of deep learning models. This applies isotropic Gaussian noise to gradients during training, which can perturb these gradients in any direction, damaging utility. Metric DP, however, can provide alternative mechanisms based on arbitrary metrics that might be more suitable for preserving utility. In this paper, we apply *directional privacy*, via a mechanism based on the von Mises-Fisher (VMF) distribution, to perturb gradients in terms of *angular distance* so that gradient direction is broadly preserved. We show that this provides both $\varepsilon$-DP and $\varepsilon d$-privacy for deep learning training, rather than the $(\varepsilon, \delta)$-privacy of the Gaussian mechanism; we observe that the $\varepsilon d$-privacy guarantee does not require a $\delta > 0$ term but degrades smoothly according to the dissimilarity of the input gradients.

As $\varepsilon$s between these different frameworks cannot be directly compared, we examine empirical privacy calibration mechanisms that go beyond previous work on empirically calibrating privacy within standard DP frameworks using membership inference attacks (MIA); we show that a combination of enhanced MIA and reconstruction attacks provides a suitable method for privacy calibration. Experiments on key datasets then indicate that the VMF mechanism can outperform the Gaussian in the utility-privacy trade-off. In particular, our experiments provide a direct comparison of privacy between the two approaches in terms of their ability to defend against reconstruction and membership inference.

## 1 Introduction

A well-known problem with machine learners is that they leak information about items used in the training set, making them vulnerable to a variety of attacks. For example, Fredrikson et al. (2015) used information about prediction confidence from machine learning APIs to show that a model inversion attack could accurately reconstruct images from a facial recognition training set; Zhu et al. (2019) showed that the same was possible using information only from gradients in the training process, for various well-known computer vision datasets; and Shokri et al. (2016) studied how training samples can be identified based on the probability scores returned in classification tasks.

One popular method for protecting against attacks is to use Differential Privacy (DP) (Dwork & Roth, 2014) in the training phase of deep neural networks, for example, applied to Stochastic Gradient Descent (SGD) in the form of DP-SGD Song et al. (2013). DP-SGD in its original form applies $\varepsilon$-DP noise to the gradient vectors within each batch, thereby providing an $\varepsilon$ guarantee over training datasets. Song et al. (2013) noted that the noise introduced by a DP mechanism impacted SGD performance significantly, but later developments have improved its performance: for example, Abadi et al. (2016) proposed a Gaussian-based mechanism with a moments accounting method for tighter bounds on the privacy budget; in the space of language models, McMahan et al. (2018) showed how to use DP for user-level privacy at the cost of increased computation rather than decreased utility; also in that space, Li et al. (2022a) showed that very different regions of the hyperparameter space relative to non-private models, and a new 'ghost clipping' technique on gradients, could lead to strong performance of large language models under DP-Adam, an extension of DP-SGD. Nevertheless, there is still generally a gap in performance between non-private and private models.

Arguably the most popular noise-adding mechanism used in DP-SGD is the Gaussian mechanism, introduced by Abadi et al. (2016) which applies Gaussian noise to gradients of deep learning models during training.

This type of noise is isotropic: that is, the generated noise vector is equally likely to point in any direction in the high-dimensional space of the deep learning model gradients. In contrast, we might expect that the utility of the model (as measured by the gradients it outputs) would be better served by a mechanism which is designed to preserve the *direction* of the gradients. Intuitively, the more the gradient directions are preserved, the better is the gradient descent algorithm going to follow the correct trajectory so as to minimise the loss function.

The idea of tuning the *shape* of the noise arises in the context of *metric differential privacy* or *d*-privacy (Chatzikokolakis et al., 2013), a generalisation of differential privacy in which the notion of adjacency is generalised to a distinguishability metric *d*. Metric differential privacy is a unifying definition which subsumes both central and local differential privacy, the former recoverable by choosing *d* to be the Hamming metric on databases, and the latter by choosing *d* to be the Discrete metric on individual data points. Importantly, by careful choice of the metric *d*, *d*-privacy mechanisms can provide a better privacy-utility trade-off than with standard differential privacy.[1]

Guided by metric DP, a natural alternative mechanism to apply to gradients is one which preserves angular distance, and hence their direction. Recently, one such *directional privacy* mechanism has been developed by Weggenmann & Kerschbaum (2021), who applied the idea to recurrent temporal data in the context of a dataset of sleep recordings. The authors provide two novel *d*-privacy mechanisms for their directional privacy, based on the von Mises-Fisher and Purkayastha distributions. The **key idea** in the present paper is to implement DP-SGD using directional noise applied to gradients, so that with high likelihood a reported gradient is close in direction to the original gradient and further away (in direction) with diminishing likelihood. The aim of the present paper is to show that these kinds of directional privacy mechanisms applied to deep learning training can have less impact on model performance because the application of noise can be more targeted while providing $\varepsilon$-privacy guarantees via metric DP.

We evaluate DP-SGD under two different regimes: the Gaussian-noise mechanism of Abadi et al. (2016), and the (directional privacy) von Mises-Fisher mechanism of Weggenmann & Kerschbaum (2021). A key question which arises when using different types of differential privacy mechanisms is *how to compare the epsilons?* since their guarantees are often not comparable, as noted in, for example, Jayaraman & Evans (2019) and Bernau et al. (2021). Motivated by the example of these other works, we analyse the suitability of two membership inference attacks (MIAs) as well as two gradient-based reconstruction attacks. Based on this analysis, we propose to compare the Gaussian and von Mises-Fisher privacy mechanisms wrt their efficacy at preventing both MIAs and reconstruction attacks while achieving some level of utility.

In summary, we define a model DIRDP-SGD and the corresponding privacy mechanism for applying directional privacy to gradients in deep learning training (§3). This mechanism comes with a metric DP guarantee, and we show that it also provides standard $\varepsilon$-DP. However, since it is not straightforward to compare $\varepsilon$-DP with the $(\varepsilon, \delta)$ guarantees of the Gaussian mechanism, we provide experimental comparisons of both privacy and utility of DIRDP-SGD and the Gaussian DP-SGD (§4), where the experimental evaluation of privacy is based on both an enhanced MIA of Hu et al. (2022) and a method that permits the reconstruction of training set data based on gradients during training (Geiping et al., 2020). In doing this we also shed light on using either kind of method alone (MIA, reconstruction) to measure privacy. We then show (§6) that DIRDP-SGD performs notably better on some major datasets for comparable levels of defence against the aforementioned attacks.

In this paper, our contributions are as follows:

- We apply for the first time a metric DP mechanism based on angular distance — via the von Mises-Fisher distribution — to use as an alternative to Gaussian noise in training via Stochastic Gradient Descent in deep learning;

- We demonstrate that this provides $\varepsilon d_\theta$-privacy (for angular distance $d_\theta$) as well as $\varepsilon$-DP for the training as a whole;

---

[1]See Chatzikokolakis et al. (2013) for an example on statistical datasets.

- We analyse both MIAs and gradient-based reconstruction attacks as candidates for empirically comparing privacy, and show why using both together is appropriate in this context.

- Given this, we show that overall on major datasets, our VMF mechanism outperforms Gaussian noise when defending against attacks.

## 2 Related Work

In this section, we review relevant work on the use of DP in deep learning (§2.1), and on metric DP, including its own intersections with deep learning (§2.2). As it is not possible to analytically compare the privacy provided by each framework, as is usually done within standard DP just by comparing values of $\varepsilon$ and $\delta$, we discuss options for an empirical evaluation of privacy. We first review membership inference attacks (§2.3), which have been used for empirical privacy comparisons within standard DP frameworks; and then, because of various issues that we identify, we also review gradient-based reconstruction attacks (§2.4).

### 2.1 Differential Privacy in Deep Learning

Neural networks can be victims of several types of attacks, like membership inference (Shokri et al., 2016; Irolla & Châtel, 2019), model stealing (Yu et al., 2020) and data reconstruction (Zhu et al., 2019; Zhao et al., 2020; Geiping et al., 2020; Wei et al., 2020). This motivates the need for privacy guarantees to protect neural networks while keeping their utility for the task they are trained to deal with.

Song et al. (2013) proposed Differentially Private Stochastic Gradient Descent (DP-SGD), which first brought DP to the training of gradient-descent models. DP-SGD adds calibrated noise in the gradients during training, before updating the parameters. This was followed by works that looked at providing efficient algorithms and tightening error bounds (Bassily et al., 2014, for example) so that the addition of noise would not degrade utility to impractical levels. A key work in this direction was made by Abadi et al. (2016), who introduced a technique to keep track of the privacy budget, called the Moments Accountant, specifically for the Gaussian mechanism.

Afterwards, several papers studied the effect of DP in deep learning in other domains, such as NLP (McMahan et al., 2018), and in applications like Generative Adversarial Networks (Xu et al., 2019; Torkzadehmahani et al., 2019). Recent work has also returned to the possibility of feasibly applying DP through output perturbations (Lu et al., 2022). The many ways in which DP has been applied in deep learning, in general, are beyond the scope of the present work, and we refer the reader to surveys such as Gong et al. (2020); below we focus only on DP-SGD and related methods.

In this context, the additional privacy comes with a cost, in that the noisy gradients may affect the utility of the model. Therefore, either better features may be collected or handcrafted, or even more data may be needed (Tramer & Boneh, 2021). Li et al. (2022a) (in NLP) and De et al. (2022) (in computer vision) also found that DP-SGD can perform well in very different regions of the hyperparameter space relative to non-private models. The architecture of the model may also play a role in the utility, with larger and pretrained models being more efficiently fine-tuned, especially with larger batch sizes (Li et al., 2022a; Anil et al., 2021), which can be computationally demanding; Li et al. (2022a) also showed how to reduce the high memory consumption for training via 'ghost clipping'. Changes to other aspects of models can also improve privacy-utility trade-offs in the use of Gaussian noise, such as using bounded activation functions like tempered sigmoids Papernot et al. (2021).

Proposals to change the DP-SGD algorithm itself have also been made, many of them relating to clipping strategies. Xu et al. (2021) observed that clipping and noise addition affect underrepresented classes, making the accuracy of the model for them even lower. Thus they proposed to control the contribution of samples in a group according to the group clipping bias. Liu et al. (2021) proposed to divide gradients from $m$ samples into $k$ groups. Before noise is added, the gradients in each group are clipped with a different bound, as opposed to a global bound from DP-SGD. They argue that a clipping could distort gradient information.

However, all these works in DP and deep learning have adopted isotropic noise, often from the Gaussian distribution. Clipping the gradients derived from these noises limits their *length*, but does not alter their *direction*. There is a lack of studies comparing how different noise distributions affect the privacy/utility

trade-off and how noise distributions other than isotropic ones can be used during the training of neural networks.

## 2.2 Metric Differential Privacy

There have been many variants of DP proposed in the literature (Pejó & Desfontaines, 2022). In this work, we adopt a relaxation of DP called metric differential privacy (hereafter metric DP), introduced by Chatzikokolakis et al. (2013) and also known as generalised DP, $d$-privacy, and $d_{\mathcal{X}}$-privacy.

Metric DP was first applied to the problem of geo-location privacy (Andrés et al., 2013) in which the user's goal is to conceal their exact location while revealing an approximate location to receive a location-based service. Many later applications of metric DP have been in this kind of geo-location context, for example, mobility tracing (Chatzikokolakis et al., 2014), location data with temporal correlations (Xiao & Xiong, 2015), mobile crowdsensing (Wang et al., 2018), and location data with non-circular distributional characteristics (Zhao et al., 2022).

In the area of deep learning in NLP, Fernandes et al. (2019) proposed a metric DP mechanism for authorship privacy using the Earth Mover's distance as the metric. Work following on from that used hyperbolic rather than Euclidean spaces for hierarchical representations (Feyisetan et al., 2019), calibrated multivariate perturbations (Feyisetan et al., 2020), representations for contextual rather than static language models (Qu et al., 2021), and a variational autoencoder to perturb overall latent vectors rather than individual words (Weggenmann et al., 2022). A related application that takes a similar spatial perspective has been to k-means clustering (Yang et al., 2022). None of these is concerned with differentially private training of a deep learner in the manner of DP-SGD.

The application of metric DP that we draw on is not related to the existing uses in deep learning just described. In the context of providing privacy guarantees to sleep study data, Weggenmann & Kerschbaum (2021) applied metric DP to periodic data noting that periodicity can be represented as a direction on a circle, and 'directional noise' perturbs this direction while preserving utility. They proposed a variety of privacy mechanisms, including variants of Laplace, plus the novel Purkayastha and von Mises-Fisher (VMF) mechanisms. Weggenmann & Kerschbaum (2021) also provided e.g. sampling methods for VMF that reduce from a multivariate sampling problem to a univariate one in order to avoid the curse of dimensionality. In the present work, we adopt the VMF mechanism to apply directional noise to gradients instead of (isotropic) Gaussian noise more typically used in DP-SGD that perturbs the gradient in any direction, drawing on a similar intuition that preserving the gradient directions should provide better utility.

## 2.3 Membership Inference Attacks

One widely discussed privacy concern in deploying deep learning models is to ensure that no training data can be recovered by a malicious user. However, Shokri et al. (2016) showed that, under certain assumptions, it is possible to infer whether a sample was used to train a model by analysing the class probabilities it outputs during inference. The attack was baptised as a membership inference attack (MIA). There is extensive literature on MIA that is beyond the scope of this paper; see for example the survey of Hu et al. (2022). Here we focus on giving a brief overview, discussing the application to calibrate levels of privacy within the standard DP framework, and noting issues that have been raised.

**Overview** Following the framework of Shokri et al. (2016), MIA works in two steps. In the first step, shadow models are trained with the aim of mimicking the target model to be attacked. In the second step, for each class, separate (binary) attack models are trained on the shadow models' prediction vectors to predict whether a sample was used to train the target or not(*in/out*).

The underlying intuition is that the distributions between *in* samples and *out* samples are different enough for the binary classifiers to identify whether the test samples belong to the target model training set. More specifically, the target model should be more confident when classifying samples it has already seen.

A number of later works extended this attack. For instance, Yeom et al. (2018) proposed an inference method that uses average training loss in its attack and requires only one query to the model target, in contrast to the large number of shadow models under the Shokri et al. (2016) approach. Salem et al. (2019) relaxed some of the assumptions of Shokri et al. (2016), such as knowledge of the target model architecture, or the

training data for target and shadow models coming from the same distribution, even though the samples are disjoint. They also reduced the number of shadow models and proposed a method that finds a threshold based on the highest posterior returned from the target's predictions with little difference in the performance of the attack. Later, Choquette-Choo et al. (2021) relaxed the constraint of having access to the prediction confidences and used only hard labels, at the expense of querying the target model several times. Recently, Ye et al. (2022) presented an enhanced MIA that they characterise as 'population-based', using reference models to achieve a significantly higher power (true positive rate) for any (false positive rate) error, at a lower computation cost. They also define an indistinguishability metric that is a function of attack AUC to more precisely characterise privacy leakage for their purpose of comparing MIAs.

Defences against MIA work in different ways. Differential privacy was mentioned by Shokri et al. (2016) as a possible defence strategy. It has been successfully adopted (Bernau et al., 2021; Choquette-Choo et al., 2021) using the Gaussian mechanism.

**Use in Calibrating $\varepsilon$ in DP**    MIA has been proposed as an empirical method of privacy assessment to help in comparing privacy budgets across DP variants.

Yeom et al. (2018) first drew out the connection between privacy risk and overfitting using MIA and attribute inference, under both their own and the original Shokri et al. (2016) approaches.

Then, Jayaraman & Evans (2019) used MIA to compare $\varepsilon$s under the original version of DP (Dwork & Roth, 2014) with those under three commonly used 'relaxed' variants — Concentrated DP (Dwork & Rothblum, 2016), Zero Concentrated DP (Bun & Steinke, 2016) and Rényi DP (Mironov, 2017). Experimentally, they used as attack frameworks both the MIA of Shokri et al. (2016) and the MIA and attribute inference of Yeom et al. (2018); CIFAR-100 (images) and Purchase-100 (customer purchase records) as datasets; and logistic regression and a two-layer Multi-Layer Perceptron (MLP) as learners. Characterising privacy leakage as the attack advantage under an MIA as per Yeom et al. (2018), where attack advantage is defined in terms of the success of an MIA, they found that for a given nominal privacy budget $\varepsilon$, privacy leakage was quite different across the various types of DP. The 'relaxed' ones experience more privacy leakage as measured by the attacks: that is, $\varepsilon$s do not have the same meaning, even across standard DP frameworks.

Bernau et al. (2021) similarly took an empirical approach to compare local and central DP, using average precision over MIAs. Across three datasets (Purchase and Texas datasets (although sampled differently from earlier work), and the Labelled Faces in the Wild (LFW) dataset representing images) and MLPs and CNNs models as appropriate, they conclude that "while the theoretic upper bound on inference risk reflected by $\varepsilon$ in LDP is higher by a factor of hundreds or even thousands in comparison to CDP, the practical protection against a white-box MI attack is actually not considerably weaker at similar model accuracy": as for Jayaraman & Evans (2019), the $\varepsilon$ values are not comparable. MIA has further been proposed for a broader assessment of privacy leakage through the ML Privacy Meter tool (Nasr et al., 2018; Kumar & Shokri, 2020; Ye et al., 2022).

One other recent approach is that of Jagielski et al. (2020), who used a novel poisoning attack, in order to audit specific levels of privacy within a DP framework.

**Issues**    Irolla & Châtel (2019) point out that MIA explores overfitting, and the larger the gap between training and evaluation accuracies, the more likely an attack would succeed. Hence, dropout, model stacking (Salem et al., 2019), as well as $l_2$ regularisation (Choquette-Choo et al., 2021) have prevented the attack by simply reducing overfitting. Moreover, Rezaei & Liu (2021) also state that results in the literature often reported only metrics (e.g. accuracy or precision) for the positive class (member class), which may be misleading because the attacks can show high false positive rates. This is problematic because in real life most samples do not belong to the member class, and thus a high positive rate would yield a very low precision in practice. The authors conclude that the high false positive rates (found even in extremely overfitted models) turn MIA impractical. They also couldn't find a good attack with both high accuracy and a low false positive rate.

Watson et al. (2022) also states that existing works present high false positive rates. Moreover, attackers may predict non-member samples incorrectly, but with high confidence, something that would be expected for member samples only. They propose enhancements to improve the attack by calibrating the membership

score to the difficulty of correctly classifying the sample, but it requires white-box access to the target model, defeating the initial concept of MIA as a black-box attack.

Given these issues, we assess the suitability of MIA for comparing privacy frameworks in our context later in this paper (§5).

### 2.4 Gradient-based Reconstruction Attacks

In light of the issues faced by MIA, we consider another attack to evaluate privacy. Reconstructions via model inversion attacks have already been demonstrated to lead to potentially serious privacy leakages in the areas of pharmacology (Fredrikson et al., 2014) and computer vision (Fredrikson et al., 2015); Fredrikson et al. (2014) already began exploring relationships with DP.

In this section, we focus on gradient-based reconstruction, as particularly relevant to DP-SGD and our variant as adding noise to gradients during training. The attack scenario is situated within a distributed learning framework. Distributed training aims to train a neural network without centralising data. It has the benefit of not having to hold private data in a single place. It consists of multiple clients, and each one holds its own private training set. Instead of sharing the data, the clients train their neural network and exchange the gradients. However, it is still possible to reconstruct the private training data from the gradients received.

The seminal study of Zhu et al. (2019) discovered that with few iterations it is possible to recover the private data in attacking neural network architectures which are twice differentiable; their attack has subsequently been referred to as the Deep Leakage from Gradients (DLG) attack. The attacker creates dummy inputs and labels, but instead of optimising the model weights, it optimises the dummy input and labels to minimise the Euclidean distance between their gradients and the gradients received from another client. Matching the gradients transforms the fake input to be similar to the real one.

This attack was refined in further works. Zhao et al. (2020) proposed iDLG (*i* stands for *improved*), which works against any differentiable network trained with cross-entropy loss over one-hot labels. The Inverting Gradients method (IGA), from Geiping et al. (2020), maximises the cosine similarity between gradients. Thus it relies on an angle-based cost function, which should be more robust than a magnitude-based one against a trained neural network (which produces gradients with smaller magnitudes). Wei et al. (2020) (baptised Client Privacy Leakage — CPL) studied how different configurations impact the effectiveness of the attack, such as different ways of initialising the dummy data.

There have been further proposed methods, such as that of Yin et al. (2021), and evaluations of multiple methods, such as that of Huang et al. (2021). In this paper, however, we focus on the earlier established methods, as we are not interested in necessarily the best performing reconstruction, but rather a method that can be used reliably in our empirical privacy calibration.

Zhu et al. (2019), in proposing DLG, also proposed some suggestions for possible defences. In addition to measures like gradient quantization and compression/sparsification, it also included the addition of Gaussian noise to gradients, although not within a DP context. Recently, Scheliga et al. (2022) proposed a variational bottleneck-based preprocessing module that aims to disguise the original latent feature space that is vulnerable to gradient-based reconstruction attacks, by learning a joint distribution between input data and latent representation. Like Zhu et al. (2019), this also does not come with differentially private guarantees.

## 3 The Privacy Model

A differential privacy mechanism can be described formally as a function that takes as input an element (drawn from a domain $\mathcal{X}$) and produces a randomised value drawn from some distribution over outputs $\mathcal{Y}$, satisfying the characteristic DP in the equation:

$$Pr(\mathcal{M}(x))[Y] \leq e^{\varepsilon} \times Pr(\mathcal{M}(x'))[Y] + \delta , \tag{1}$$

for all $x \sim x' \in \mathcal{X}$ and $Y \subseteq \mathcal{Y}$ and where $\varepsilon > 0$ and $0 \leq \delta < 1$. When $\delta = 0$ then Eqn (1) is called *pure $\varepsilon$-DP*; otherwise it is called *approximate*-DP or simply $(\varepsilon, \delta)$-DP.

Popular methods of randomisation include the Gaussian, the Laplace (when the outputs are continuous) or the Geometric (when the outputs are discrete), all of which involve the addition of noise to the input $x \in \mathcal{X}$ to produce the noisy output $y \in \mathcal{Y}$.

Metric DP is a generalisation of pure $\varepsilon$-DP in which the adjacency relation $\sim$ is replaced with a distinguishability metric $d$.

**Definition 1** *(Metric differential privacy) (Chatzikokolakis et al., 2013) Let $\varepsilon > 0$. A mechanism $\mathcal{M}$ on an (input) metric space $(S, d)$, where $S$ is a set and $d$ is a metric, and producing outputs over $\mathcal{Z}$, satisfies $\varepsilon d$-privacy, if for all $s, s' \in S$ and $Z \subseteq \mathcal{Z}$,*

$$Pr(\mathcal{M}(s))[Z] \leq e^{\varepsilon d(s,s')} \times Pr(\mathcal{M}(s'))[Z] \ ,$$

*where $Pr(\mathcal{M}(s))[Z]$ means the probability that the output of applying mechanism $\mathcal{M}$ to $s$ lies in $Z$.*

Def 1 says that when two inputs $s, s'$ differ by the amount $d(s, s')$, the mechanism can make them indistinguishable up to a ratio proportional to $e^{\varepsilon d(s,s')}$. This means that when points are farther apart they become easier to distinguish.

Metric DP, like pure $\varepsilon$-DP, has straightforward composition properties, [2] and satisfies the data processing inequality (Chatzikokolakis et al., 2013). Approximate-DP, however, requires approximate methods for computing tight bounds on $(\varepsilon, \delta)$ under composition.

Our application to deep learning uses a metric based on angular distance of vectors, which we describe in the next sections.

### 3.1 Standard DP-SGD

The standard DP-SGD from Abadi et al. (2016) is shown in Algorithm 1. It differs from the original stochastic gradient descent (i.e. without perturbation) only at lines 10 and 13, where the gradients $g_t(x_i)$ are first clipped and then perturbed using the Gaussian distribution. This is implemented essentially by adding a random perturbation to each of the components of the gradient when represented as a point in $\mathbb{R}^K$.

---

**Algorithm 1** DP-SGD with Gaussian noise

---

1: **Input:** Examples $\{x_1, \ldots, x_N\}$, loss function $\mathcal{L}(\theta) = \frac{1}{N} \sum_i \mathcal{L}(\theta, x_i)$. Parameters: learning rate $\eta_t$, noise scale $\sigma$, group size $L$, gradient norm bound $C$.
2: **Initialise** $\theta_0$ randomly
3: **for** $t \in T$ **do**
4:     **Take a random batch**
5:     $L_t \leftarrow$ random sample of $L$ indices from $1 \ldots N$
6:     **for** $i \in L_t$ **do**
7:         **Compute gradient vector**
8:         $\mathbf{g}_t(x_i) \leftarrow \nabla_{\theta_t} \mathcal{L}(\theta_t, x_i)$
9:         **Clip gradient vector**
10:        $\overline{\mathbf{g}}_t(x_i) \leftarrow \mathbf{g}_t(x_i) / \max(1, \frac{\|\mathbf{g}_t(x_i)\|_2}{C})$
11:     **end for**
12:     **Add noise**
13:     $\tilde{\mathbf{g}}_t \leftarrow \frac{1}{L} \sum_i (\overline{\mathbf{g}}_t(x_i) + \mathcal{N}(0, C^2\sigma^2))$
14:     **Descent**
15:     $\theta_{t+1} \leftarrow \theta_t - \eta_t \tilde{\mathbf{g}}_t$
16: **end for**
17: **Output** $\theta_T$

---

The Gaussian mechanism satisfies approximate-DP, and therefore Abadi et al. developed the moments accounting method for improved composition bounds. We adopt this method for evaluating $(\varepsilon, \delta)$ in our experiments using the Gaussian in DP-SGD.

---

[2]That is, the epsilons "add up" under sequential composition, and for the same metric, and privacy does not diminish under parallel composition.

## 3.2 Directional Privacy and DirDP-SGD

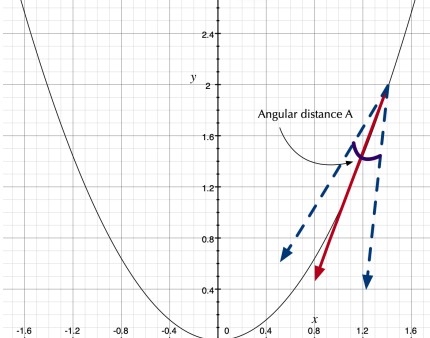

How the gradients are perturbed during the DP-SGD. The red line is the unperturbed gradient, and the dotted blue lines are perturbations of angular distance $A$.

Figure 1: Perturbed gradients

Gradient descent optimises the search for parameter selection that minimises the loss. Thus an alternative method of perturbing the gradients is to use randomisation that is based on perturbing the angle of deviation from the original gradient. To give some intuition, Figure 1 illustrates how a gradient of a convex curve can be perturbed, leading to a perturbation of the descents.

Given two vectors $v, v'$ in $\mathbb{R}^K$, we define the angular distance between them as $d_\theta(v, v') = \frac{\arccos v^T v'}{\|v\|\|v'\|}$. When $v, v'$ are, for example, vectors on the unit $K$-dimensional sphere, then $d_\theta$ becomes a metric. Following Weggenmann & Kerschbaum (2021), we can use this to define *directional privacy*.

**Definition 2** *(Directional Privacy) (Weggenmann & Kerschbaum, 2021) Let $\varepsilon > 0$. A mechanism $\mathcal{M}$ on $\mathbb{R}^K$ satisfies $\varepsilon d_\theta$-privacy, if for all $v, v'$ and $Z \subseteq supp\mathcal{M}$,*

$$Pr(\mathcal{M}(v))[Z] \leq e^{\varepsilon d_\theta(v,v')} \times Pr(\mathcal{M}(v'))[Z] \ .$$

Definition 2 says that when the mechanism $\mathcal{M}$ perturbs the vectors $v, v'$, the probabilities that the perturbed vectors lie within a (measurable) set $Z$ differ by a factor of $e^{\varepsilon d_\theta(v,v')}$. This means that the smaller the angular distance between the vectors $v, v'$ the more indistinguishable they will be.

Weggenmann & Kerschbaum (2021) introduced the von Mises-Fisher (VMF) mechanism derived from the VMF distribution that perturbs an input vector $x$ on the $K$-dimensional unit sphere.

**Definition 3** *The VMF mechanism on the $K$-dimensional unit sphere is given by the density function:*

$$\mathcal{V}(\varepsilon, x)(y) \ = \ C_K(\varepsilon)e^{\varepsilon x^T y} \ ,$$

*where $\varepsilon > 0$ and $C_K(\varepsilon)$ is the normalisation factor.*

The authors proved that the VMF mechanism satisfies $\varepsilon d_\theta$-privacy.

They also provide an analysis of the expected error of the VMF[3] as well as sampling methods which we use later in our experiments.

Importantly, the following also holds.

**Theorem 1 (Weggenmann & Kerschbaum (2021))** *Let $\varepsilon > 0$ and denote by $\mathbb{S}^{K-1}$ the unit sphere in $K$ dimensions. Then the VMF mechanism on $\mathbb{S}^{K-1}$ satisfies $\varepsilon d_2$-privacy where $d_2$ is the Euclidean metric. That is,*

$$\mathcal{V}(\varepsilon, x)(Y) \ \leq e^{\varepsilon d_2(x,x')}\mathcal{V}(\varepsilon, x')(Y) \ ,$$

*for all $x, x' \in \mathbb{S}^{K-1}$ and all (measurable) $Y \subseteq \mathbb{S}^{K-1}$.*

---

[3]See Weggenmann & Kerschbaum (2021) for full details.

In our application, we will need to apply the basic mechanism $\mathcal{V}$ to more complex data representations, namely where a point is a represented as convex sum of $m$ orthogonal vectors in $n$-dimensions. The standard method for doing this is to apply $m$-independent applications of the mechanism (in this case $\mathcal{V}$); the differential privacy guarantee is then parametrised by $m$ as follows.

**Corollary 1 (Dwork & Roth (2014))** *Let $\mathcal{V}$ be the mechanism defined in Definition 3. Let $v, v'$ be two vectors on the unit sphere, where $v = \lambda_1 u_1 + \ldots \lambda_k u_m$ and $v' = \lambda'_1 u'_1 + \ldots \lambda'_k u'_m$, where $u_i, u'_i$ for $1 \leq i \leq m$ are vectors on the unit sphere, and $|\lambda_i|, |\lambda'_i| \leq 1$. Define $\mathcal{V}^*$ to be the mechanism that applies $\mathcal{V}$ independently to each of the $u_i/u'_i$ to produce random vectors distributed respectively as: $\mathcal{V}^*(\varepsilon, v), \mathcal{V}^*(\varepsilon, v')$. Then*

$$\mathcal{V}^*(\varepsilon, v)(Y) \leq e^{2\varepsilon\sqrt{m}} \mathcal{V}(\varepsilon, v')(Y) .$$

**Proof:** *The standard properties of differential privacy (Dwork & Roth, 2014) result in the following relationship:*

$$\mathcal{V}^*(\varepsilon, v)(Y) \leq e^{\varepsilon \sum_{1 \leq i \leq i'} d_2(\lambda u_i, \lambda' u'_i)} \mathcal{V}(\varepsilon, v')(Y) .$$

*Observe that for any orthonormal set of vectors $u_i$ on the unit sphere, we have that $\sum_{0 \leq i \leq m} d_2(0, \lambda_i u_i) \leq \sum_{0 \leq i \leq m} |\lambda_i|\sqrt{m}$. The result now follows using the triangle inequality of $d_2$, and that $d_2(\lambda_i u_i, \lambda'_i u'_i) \leq 2$.* $\square$

There are a number of interesting scenarios based on Corollary 1 which we will explore in our adaptation of DP-SGD below. The first is that $\mathcal{V}$ is applied (once) to an $n$-dimensional vector to produce a random $n$-dimensional vector. For us, in Corollary 1 we would use $m = 1$ to obtain an overall $2\varepsilon$ for our privacy parameter. An alternative extreme is to apply noise independently to each of the components (in the way that the original DP-SGD does); Corollary 1 then gives a $\varepsilon\sqrt{n}$ privacy budget. An interesting hybrid scenario, not available for the Gaussian distribution but available for the $\mathcal{V}$ mechanism, is to partition the $n$-dimensional components into $m$-orthogonal components and to apply $\mathcal{V}$ independently to each of those components; in this case, we obtain the $\varepsilon\sqrt{m}$ privacy budget as in Corollary 1. As explained in our experimental sections below, we found that for some of the datasets, this was a useful generalisation for the purposes of efficiency.

Using Def 3 we can now design a new DP-SGD algorithm using the VMF mechanism which perturbs the *directions* of the gradients computed in SGD. This algorithm, which we call DIRDP-SGD, is depicted in Algorithm 2.

---

**Algorithm 2** DIRDP-SGD with von Mises-Fisher noise

---

1: **Input:** Examples $\{x_1, \ldots, x_N\}$, loss function $\mathcal{L}(\theta) = \frac{1}{N} \sum_i \mathcal{L}(\theta, x_i)$. Parameters: learning rate $\eta_t$, noise scale $\sigma$, group size $L$, gradient norm bound $C = 1$.
2: **Initialise** $\theta_0$ randomly
3: **for** $t \in T$ **do**
4:     **Take a random batch**
5:     $L_t \leftarrow$ random sample of $L$ indices from $1 \ldots N$
6:     **for** $i \in L_t$ **do**
7:         **Compute gradient vector**
8:         $\mathbf{g}_t(x_i) \leftarrow \nabla_{\theta_t} \mathcal{L}(\theta_t, x_i)$
9:         **Scale gradient vector**
10:        $\overline{\mathbf{g}}_t(x_i) \leftarrow \mathbf{g}_t(x_i) / \frac{\|\mathbf{g}_t(x_i)\|_2}{C}$
11:     **end for**
12:     **Add noise**
13:     $\tilde{\mathbf{g}}_t \leftarrow \frac{1}{L} \sum_i \mathcal{V}(\sigma, \overline{\mathbf{g}}_t(x_i))$
14:     **Descent**
15:     $\theta_{t+1} \leftarrow \theta_t - \eta_t \tilde{\mathbf{g}}_t$
16: **end for**
17: **Output** $\theta_T$

---

In order to prove a DP guarantee, Algorithm 2 is modified from the original DP-SGD in three ways. First, we fix $C$ to 1. Then, in line 10, instead of clipping the gradients, we scale the gradients to the clip length

(ie. 1). Finally, in line 13, instead of adding a noisy vector to the gradient, we generate a noisy gradient directly using the VMF mechanism.[4]

We now show Algorithm 2 satisfies both $\varepsilon$-DP and $\varepsilon d_\theta$-privacy in terms of indistinguishability of batches used in the learning, viz that if two batches (composed of data points $x_i$) differ in only one point, then they are probabilistically indistinguishable.

**Theorem 2** *Denote by $B = [v_1, \ldots v_n]$ , and $B' = [v'_1, \ldots v'_n]$ two batches of vectors (gradients). If batch $B'$ differs from $B$ in at most one component vector, then Algorithm 2 satisfies $\sigma d_2$-privacy wrt. batches, namely that:*

$$Pr(\mathcal{VM}(B) \in Z) \leq Pr(\mathcal{VM}(B') \in Z) \times e^{\sigma d_2(B,B')} , \tag{2}$$

*where $Z$ is a (measurable) set of vectors, $Pr(\mathcal{VM}(B) \in Z)$ is the probability that the output vector lies in $Z$ and (abusing notation) $d_2(B, B') = \max_{B \sim B'} d_2(v, v')$ is the maximum Euclidean distance between all pairs $v \in B, v' \in B'$.*

**Proof:** *Line 10 of Algorithm 2 ensures each vector in $B, B'$ lies on the unit sphere (since $C = 1$) and line 13 applies VMF parametrised by $\sigma$. Applying Thm 1 to every vector in $B, B'$ yields Eqn (2) since the (parallel) composition of $d_2$-privacy mechanisms gives a guarantee with $Pr(\mathcal{VM}(B) \in Z) \leq Pr(\mathcal{VM}(B') \in Z) \times e^{\sigma \sum_{1 \leq i \leq n} d_2(v_i, v'_i)}$; this reduces to Eqn (2) since all but one of the distances $d_2(v_i, v'_i) = 0$, by assumption. The averaging in line 13 is an example of a post-processing step which, by the data processing inequality property of d-privacy does not decrease the privacy guarantee (Fernandes et al., 2022).*

$\square$

Observe that Algorithm 2 assumes we apply the $\mathcal{V}$ mechanism to the whole gradient; as mentioned above, in our experiments we sometimes partition the $n$-dimensional space. We proceed though to prove a privacy guarantee assuming Algorithm 2 applies $\mathcal{V}$ without partitioning.

**Corollary 2** *Algorithm 2 satisfies $\varepsilon$-DP wrt adjacent training sets $D, D'$.*

**Proof:** *Observe that $\max_{B \sim B'} d_2(v, v') = 2$ since $v, v'$ lie on the unit sphere. $\varepsilon$-DP on batches follows from choosing $\sigma = \frac{\varepsilon}{2}$ which is the standard DP-SGD tuning from Song et al. (2013). Since batches are disjoint, the result follows by parallel composition for adjacent training sets $D, D'$.* $\square$

**Corollary 3** *Algorithm 2 satisfies $\varepsilon d_\theta$-privacy.*

**Proof:** *Follows from the fact that $d_2$-privacy implies $d_\theta$-privacy (since $d_2 \leq d_\theta$ pointwise on the unit sphere), using $d_\theta$ reasoning in Thm 2 and using the same $\sigma$ tuning as per Cor 2.* $\square$

Note that the epsilons in Cor 2 and Cor 3 are not comparable as they represent different notions of privacy.

We remark that by *scaling* rather than clipping the gradients, we also protect against privacy leaks caused when the gradient length is less than $C$. (Information may or may not be leaked by knowing the length of the gradient, but by scaling rather than clipping the gradient, we remove this possibility.)

### 3.3 Notion of Theoretical Guarantees and Comparison in Practice

At this point, it is not clear how directly we can compare the two privacy guarantees for Algorithm 1 and Algorithm 2. As mentioned above the guarantee for Algorithm 1 includes a $\delta > 0$ parameter — this means that there is a risk that the perturbation will leak more than for an $\varepsilon$-private mechanism, and therefore may provide reduced protection against a threat of membership inference or reconstruction. Moreover, previous work (Chatzikokolakis et al., 2019) has shown that comparing epsilons between different privacy definitions can be problematic, and not informative. To avoid confusion we use $\varepsilon_G$ for the privacy parameter used for Algorithm 1 and $\varepsilon_V$ for Algorithm 2.

---

[4]This is because the VMF mechanism generates a new noisy vector based on its input.

We, therefore, consider empirical privacy comparisons: Membership Inference Attacks (§2.3) and gradient-based reconstruction attacks (§2.4). The former has been used in comparisons of $\varepsilon$s within standard DP; the latter is a privacy vulnerability directly linked to the key use of gradients in our deep learning training. We consider these in our particular context in §5. We simultaneously empirically compare each mechanism's utility on a classification task. Although Algorithm 1 has been widely used, Algorithm 2 is a novel application of the VMF mechanism, and one of our tasks (detailed below) is to determine ranges of the parameter $\varepsilon_V$ that provide a good trade-off between defending against the threat of privacy leakage versus allowing a set of parameters to be determined that provides an acceptable level of utility.

### 3.4 Implementing Directional Privacy

We use Opacus,[5] introduced by Yousefpour et al. (2021), as a starting point for the experiments. The library, based on PyTorch (Paszke et al., 2019), implements DP-SGD.

From an implementation view, there are three main components: (i) the minibatches are built by using Poisson sampling: each sample from the training dataset is chosen with a certain probability $p$, which means that a sample may appear in zero, or more than one times in an epoch; (ii) the sample gradients are capped to avoid a very large individual contribution from one sample; (iii) noise is added to the gradients. Only Gaussian noise is supported.

We extend Opacus to work with the VMF distribution. Component (i) is left unchanged. For component (ii), we cap gradients, bounding them by $C = 1$, to ensure $\varepsilon$-DP of Algorithm 2 (see Thm 2); for a fair comparison, we do the same for the Gaussian mechanism. For component (iii), we switch the Gaussian noise for the Von Mises-Fisher one. To sample from the VMF distribution we follow the method from Ulrich (1984) and Wood (1994).[6]. More specifically, the gradients are clipped per layer by Opacus, by flattening the its rows into a single vector. Then, for each layer, we add VMF noise per row. For our empirical evaluation, we add noise to the budget of $\varepsilon_V$ in each update.

## 4 Overall Experimental Setup

Most works evaluating DP in deep learning report performance on some task (typically, classification accuracy) for utility, but for the level of privacy, they report only on the value of $\varepsilon$ (and $\delta$ if relevant). As we note in §3.3, it is not meaningful to compare epsilons across pure-DP and approximate-DP. We, therefore, take an empirical approach to calibrating the respective epsilons, $\varepsilon_G$ and $\varepsilon_V$: we measure how DIRDP-SGD performs to prevent membership inference attacks (MIA) and compare its success against gradient-based reconstruction attacks. We empirically investigate and justify our choices in §5.

For utility, as is typically done, we compare the accuracy of different neural networks in the task of classification when they are trained with DP guarantees against the baseline without privacy guarantees.

### 4.1 DirDP-SGD: $\varepsilon_V$

Unlike Gaussian noise (§4.3), there is no prior work with VMF to use as a guide for selecting an appropriate $\varepsilon_V$. Based on preliminary experiments, we found a range of changes to utility in $\varepsilon_V \in \{1, 5, 10, 50, 500\}$; we also included $\varepsilon_V = 300,000$, which hardly shifts gradients, to investigate the effects of negligible noise.

### 4.2 Datasets

We use classification tasks from the image processing domain, as in many works, and specifically datasets used in other privacy work.[7]

- **Fashion-MNIST**[8] (Xiao et al., 2017), inspired by the MNIST dataset (Deng, 2012), contains 70,000 images of fashion products from 10 classes. The images are 28×28 pixels in greyscale. The training set contains 60,000 instances and the test set has 10,000 instances.

---

[5] https://opacus.ai

[6] https://github.com/dlwhittenbury/von-Mises-Sampling

[7] For example, Abadi et al. (2016) use two datasets, MNIST and CIFAR-10, in proposing the moments accountant for DP-SGD; Zhu et al. (2019) use MNIST, CIFAR-100, SVHN and LFW in evaluating their DLG reconstruction attack; Papernot et al. (2021) use MNIST, Fashion-MNIST and CIFAR-10 in exploring tempered sigmoid activations for deep learning with DP.

[8] https://github.com/zalandoresearch/fashion-mnist

- **CIFAR**[9] dataset (Krizhevsky, 2009) contains 60,000 coloured images of 32x32 pixels for each one of the 3 channels. It has two versions: CIFAR10, in which each image belongs to one out of 10 classes, and CIFAR100, which contains 100 classes. The training set contains 50,000 instances and the test set has 10,000 instances.

- **LFW**[10], or Labeled Faces in the Wild dataset (Huang et al., 2007), has 13,233 images of 5,749 people collected from the internet. It is a particularly interesting dataset because it is composed of people's faces, which is something that one may wish to hide to preserve their identity, and consequently has been the focus of previous high-profile work on privacy leakage (Fredrikson et al., 2015, for example). The images have 250x250 pixels, some in greyscale but most are coloured. The standard task, which we also adopt, is identity recognition; the standard training and test sets for this contain 9,525 and 3,708 instances respectively. Given its large number of classes, many with few instances, we follow Wei et al. (2020) to downsize the dataset.

  In doing this, we kept only the classes that contain at least 14 objects. This reduced the number of classes to 106 and the number of samples to 3,737 (Wei et al., 2020). Even after this, there is a strong imbalance amongst the classes, with some having dozens of members but others having hundreds. Therefore we under-sampled the majority classes by randomly picking objects so that all classes end up with 14 samples. This reduced the dataset even further, to 14*106 = 1,484 instances. Finally, we split the resulting dataset into training (80%, or 1,113 samples) and test (20%, or 371 samples) sets.

Commenting on their use of Fashion-MNIST and CIFAR-10 in a similar context to ours in terms of exploring DP, Papernot et al. (2021) note that, although these datasets are considered largely 'solved' in the computer vision community, "achieving high utility with strong privacy guarantees remains difficult", and so are suitable for comparing variants of DP, as they also do.

### 4.3 Primary Baselines

In terms of deep learning architectures to investigate, we broadly follow the setup of Scheliga et al. (2022). The architectures of neural networks we use are **LeNet**, the original convolutional neural network (CNN) proposed by Lecun (1989), and a simple Multilayer Perceptron (**MLP**) with 2 layers, which are feedforward neural networks (Goodfellow et al., 2016), in line with other work defining new approaches to privacy.[11] Scheliga et al. (2022) include MLPs as they note that Geiping et al. (2020) provide a theoretical proof that in fully connected networks, their IGA attack can uniquely reconstruct the input to the network from the network's gradients. LeNet is a prototypical architecture for CNNs.

The two baselines in terms of privacy protection for these architectures are (i) the state-of-the-art DP-SGD using **Gaussian noise** and (ii) the neural networks without any DP guarantees. We compare their performance in terms of accuracy and susceptibility to attacks against our DIRDP-SGD.

For the Gaussian noise, there are no standard guidelines on the range to test, as $\varepsilon$ does not have an easily interpretable meaning, with no universally agreed-upon point for what counts as 'too large'; Dwork et al. (2019) note that "while all small $\varepsilon$ are alike, each large $\varepsilon$ is large after its own fashion, making it difficult to reason about them." As a common range for all of our baseline/dataset combinations, we consequently select values going from 'small' ($\varepsilon_G \leq 1$) to the common largest value of 8 that a number of works (Abadi et al., 2016; De et al., 2022, for example) have selected over the years. We also added larger $\varepsilon_G = 50$ as a relatively small amount of noise, although this is larger than in many other works, for calibration purposes.

### 4.4 Evaluation Measures

We evaluate DIRDP-SGD in terms of utility and privacy. For utility, we measure the impact that different DP strategies have on the accuracy of different neural network architectures on classification tasks over different datasets. We evaluate the impact of the privacy mechanism on model performance according to

---

[9]https://www.cs.toronto.edu/~kriz/cifar.html

[10]http://vis-www.cs.umass.edu/lfw/

[11]For example, Abadi et al. (2016) use an MLP in proposing the moments accountant for DP-SGD; Zhu et al. (2019) similarly use an MLP in calibrating $\varepsilon$ across standard DP frameworks; Papernot et al. (2021) use a LeNet-style architecture in exploring tempered sigmoid activations for deep learning with DP.

different values of $\varepsilon$ and the absence of privacy guarantees. This is in line with previous works (Abadi et al., 2016; Li et al., 2022a).

As we observe earlier, the LFW dataset and associated tasks are particularly challenging relative to the other two, due to the higher number of classes and a smaller number of instances. Therefore, we might expect low accuracies, which the added noise might reduce to near-zero levels, obscuring differences among different types and levels of noise. For this dataset, then, in addition to standard accuracy, we report the Top-10 accuracy rates, where a prediction is correct if the true label is amongst any of the model's top 10 classes with the highest confidence rate (so standard accuracy is the same as Top-1 accuracy).

For the level of privacy achieved, in §5 we evaluate membership inference and reconstruction attacks as mechanisms for empirical privacy comparison and then used the selected alternative to assess how DIRDP-SGD performs compared to baselines.

## 5 Empirical Privacy Evaluation

In this section, we consider both MIA and gradient reconstruction attacks (2) for calibration of $\varepsilon$ values in our context of comparing standard and metric DP. We assess them on standard DP with Gaussian noise within our experimental setup (4).

It is well known that defining an operational interpretation of $\varepsilon$ is a challenge, with the meaning of $\varepsilon$ being contextually dependent (Lee & Clifton, 2011; Dwork et al., 2019). In line with other work proposing empirical comparisons of privacy Jayaraman & Evans (2019), we, therefore, choose an approach that directly relates to the one taken by DP-SGD and our DIRDP-SGD, which obfuscates gradients: (i) we measure how DIRDP-SGD performs to prevent membership inference attacks (MIA). We follow the framework from Shokri et al. (2016), where the goal is to identify if a sample was used to train a target model (§5.1); we also look at the more recent enhanced MIA of Ye et al. (2022) (§5.2). Then, we (ii) compare their success against gradient-based reconstruction attacks (Zhu et al., 2019; Geiping et al., 2020) (§5.3). In these attacks, the goal is to reconstruct images solely from their gradients. Obfuscating gradients successfully should to some extent then prevent this reconstruction. (Zhu et al. (2019) do this in proposing several non-DP defences against their own attack.)

At the end of the section, we draw a conclusion about which to use for our principal results (§6) comparing standard and metric DP.

### 5.1 Membership Inference Attack

MIA has become a standard benchmark for evaluating privacy. We explore first the framework proposed by Shokri et al. (2016). We analyse how a range of $\varepsilon$ affects the performance of the attack, which will serve as starting point for calibrating the privacy-utility trade-off.

#### 5.1.1 Experimental Setup

Here we expand on the high-level view of §2.3 to give some more technical detail and notation on membership inference attacks. Then, we bring our results.

**Technical Detail on MIA** Following the notation from Shokri et al. (2016), the first step of the two-step process trains several shadow models $f_{shadow}^i()$ whose job is to mimic the target model $f_{target}()$ to be attacked. The shadow models should be trained similarly to the target (i.e. same or close architecture, hyperparameters and task). Thus, they receive samples and outputs probabilities to which class each sample belongs. However, it is assumed that the training sets of $f_{shadow}^i()$ and $f_{target}()$ are disjoint, i.e., $D_{shadow}^{train} \cap D_{target}^{train} = \emptyset$. The second step is to train attack models whose job is to predict whether a sample was used to train the target or not. One attack model $f_{attack}^j()$ is trained per class. The attackers can be any binary classifier. The outputs of $f_{shadow}^i()$ (and $f_{target}()$) are probability vectors $\boldsymbol{y}$ of size $c$, being $c$ the number of possible classes. Thus, $D_{attack}^{train}$ is a collection of samples $((\boldsymbol{y}, y), m)$, where $\boldsymbol{y}$ are the outputs of the shadow models for samples $\boldsymbol{x}_{shadow}$, $y$ is the true class of $\boldsymbol{x}_{shadow}$, and $m$ indicates whether $\boldsymbol{x}_{shadow}$ was a member in $D_{shadow}^{train}$ or not. Finally, at inference time, a prediction vector $\boldsymbol{y} = f_{target}(\boldsymbol{x})$ is passed through the corresponding $f_{attack}^j()$ to asses whether $\boldsymbol{x} \in D_{target}^{train}$.

Table 1: Accuracy under different DP settings for the Membership Inference Attacks using Kulynych & Yaghini (2018). We also inform the gap between train and evaluation accuracy. A negative value means that the training accuracy was smaller than the evaluation accuracy.

| | | | F-MNIST | | CIFAR10 | |
| Model | Mechanism | $\varepsilon_G$ | Attack acc. | Train gap | Attack acc. | Train gap |
|---|---|---|---|---|---|---|
| LeNet | – | – | 50.1 | -1.1 | 49.8 | 8.9 |
| MLP | – | – | 49.8 | 4.2 | 53.5 | 34.1 |
| LeNet | Gauss | 0.8 | 50.3 | -1.3 | 49.8 | -0.5 |
| | | 8.0 | 50.2 | -0.2 | 49.7 | -0.5 |
| | | 50.0 | 50.2 | -1.3 | 50.3 | 2.0 |
| MLP | Gauss | 0.8 | 50.3 | -1.1 | 50.1 | 1.0 |
| | | 8.0 | 50.7 | -1.2 | 50.4 | 2.9 |
| | | 50.0 | 50.8 | -1.4 | 50.4 | 4.2 |

**Our Setup**   We design the MIA according to the procedure explained above. Both $f^i_{shadow}()$ and $f_{target}()$ have the same architecture: LeNet and MLP. We train 10 shadow models, and we use the Gradient Boosting Decision Tree (LightGBM) (Ke et al., 2017) as $f_{attack}()$.

Because of the LFW dataset's small size and consequent infeasibility of training multiple shadow models, we do not use it for MIA; we only apply this to the other two datasets.

For each dataset, $D$, the test set $D_{test}$ is kept intact, and we split the training set $D_{train}$ in half, one for $D_{shadow}$ and another for $D_{target}$. Both datasets are in turn split in half into training and evaluation set each, into $D^{train}_{shadow}$, $D^{train}_{target}$, $D^{eval}_{shadow}$ and $D^{eval}_{target}$. Thus, we ensure that the data are disjoint but come from the same distribution. Each $f^i_{shadow}()$ is trained with 70% of $D^{train}_{shadow}$ and $D^{eval}_{shadow}$, sampled randomly. The *in*-samples and *out*-samples used to train $f^j_{attack}()$ come from $D^{train}_{shadow}$ and $D^{eval}_{shadow}$ respectively. Even though not explicitly mentioned by Shokri et al. (2016), but following Irolla & Châtel (2019), we train $f^j_{attack}()$ only with samples correctly classified by $f^i_{shadow}()$.

For inference, we take the held-out $D_{test}$ as our *out*-samples set and randomly sample the same amount of records from $D^{train}_{target}$ to build our *in*-samples set $D_{in\_samples}$. Our final test set is their concatenation $D'_{test} = D_{test} \bigcup D_{in\_samples}$. We use the implementation provided by Kulynych & Yaghini (2018), a library published with the goal of enabling the running of MIAs against machine learning models.

### 5.1.2   Results and Discussion

Table 1 shows the accuracy of the attacks as well as the gap between training and evaluation for both private and non-private targets. The gap refers to the accuracy of the classification task, not the attack. A negative value in the gap column means that the training accuracy was smaller than the evaluation accuracy. Note that the gap refers to the accuracy of the classification task, not the attack.

Focusing first on the attacks against non-private models, it can be seen that the attacks are largely unsuccessful. The most successful is against the non-private MLP, which is only at 53%. The attack is at a random chance for non-private LeNet and all of the models with noise.

We observe the MIA marginally succeeds only when the target model is heavily overfitted and thus reports a big gap between its training and evaluation accuracies. This is consistent with several other studies in the literature, which tackle MIA with regularisation techniques (Shokri et al., 2016; Salem et al., 2019; Choquette-Choo et al., 2021). Irolla & Châtel (2019), for instance, report an attack accuracy for CIFAR10 of 56.54% when their model shows a train-test gap of 32%. For Fashion-MNIST, when the gap is 11%, their attack achieves an accuracy of only 50.92%.

Given that attacks on non-private models are unsuccessful, we would not expect attack success rates against models with protective noise added to be successful either, and consequently not useful for distinguishing among $\varepsilon$s. In Table 1 we give a small, medium and large $\varepsilon_G$ to show that this is the case.

Table 2: AUC attack scores under different DP settings for the Enhanced Membership Inference Attack using the ML Privacy Tool with reference models (MIA-**R**). We also inform the gap between train and evaluation accuracy. Negative values mean that the training accuracy was smaller than the evaluation accuracy.

| | | | **F-MNIST** | | **CIFAR10** | |
|---|---|---|---|---|---|---|
| **Model** | **Mechanism** | $\varepsilon_G$ | **Attack AUC** | **Train gap** | **Attack AUC** | **Train gap** |
| LeNet | – | – | 54.8 | 4.1 | 78.5 | 49.2 |
| MLP | – | – | 61.1 | 13.0 | 82.0 | 35.1 |
| LeNet | Gauss | 1.0 | 49.3 | 0.0 | 51.1 | 3.3 |
| | | 8.0 | 50.5 | 0.7 | 51.6 | 1.5 |
| | | 50.0 | 50.4 | 1.2 | 51.7 | 3.0 |
| | | 200.00 | 50.5 | 0.9 | 52.2 | 2.0 |
| | | 300k | 51.6 | 0.6 | 58.0 | 8.6 |
| MLP | Gauss | 1.0 | 49.9 | 3.5 | 50.5 | -0.4 |
| | | 8.0 | 51.8 | 3.2 | 52.0 | 2.2 |
| | | 50.0 | 52.6 | 3.5 | 53.9 | 4.6 |
| | | 200.0 | 53.4 | 3.7 | 56.3 | 4.8 |
| | | 300k | 56.1 | 5.2 | 70.9 | 22.4 |

## 5.2 Enhanced Membership Inference Attack

The Enhanced Membership Inference Attack Ye et al. (2022) has not yet been used for the same kinds of calibration of $\varepsilon$ as the older MIA above. In this section, we explore it in the same manner as in §5.1.

### 5.2.1 Experimental Setup

**Technical Detail on Enhanced MIA via Reference Models (MIA-R)** Ye et al. (2022) propose enhancements for MIA to achieve a higher true positive rate for any false positive rate. In the MIA via **R**eference models attack (called Attack R in their paper; from now on we will refer to it as MIA-**R**) trains several reference models (akin to shadow models) with $D_{refs}^{train}$. Later, the attack must find whether $(\boldsymbol{x}_z, \boldsymbol{y}_z)$ belongs to $D_{target}^{train}$ or $D_{target}^{test}$. It does so by comparing the loss of the target model against a threshold function that depends on the target data feature and label:

$$\ell(\theta, \boldsymbol{x}_z, \boldsymbol{y}_z) \leq c_\alpha(\boldsymbol{x}_z, \boldsymbol{y}_z)$$

where $c_\alpha(.)$ is the threshold function satisfying an arbitrary confidence $0 \leq \alpha \leq 1$ and $\theta$ is the target model parameters.

**Our Setup** We use the ML Privacy Meter[12] (Shokri et al., 2016; Nasr et al., 2018; Kumar & Shokri, 2020; Ye et al., 2022) for the implementation, which is designed to enable a consistent framework for evaluating leakage from MIAs. We train 10 reference models, which follow the same architecture and hyperparameters as the target models, except that the reference models are never trained with DP.

We let $D_{target}^{train}$ and each one of the ten splits of $D_{refs}^{train}$ to have 4,500 images for CIFAR10 and 5,000 images for Fashion-MNIST. For both datasets, $D_{target}^{test}$ has 900 images. To evaluate the attack, our $D_{in} = D_{target}^{train}$ and $D_{out} = D_{target}^{test}$, and thus the final test set is $D_{test} = D_{in} \bigcup D_{out}$.

The success of the attack is quantified by a ROC curve, which translates to a trade-off between False Positive Rate (FPR, or classifying unseen samples as training samples) and True Positive Rate (TPR, or correctly identifying training samples). The curve is summarised by a single number, its Area Under the Curve (AUC), and can be seen as the aggregate privacy risk to the data leaked by the model under attack (Kumar & Shokri, 2020). The more successful the attack is, the bigger the TPR will be at a small FPR leading to a bigger AUC. We use this AUC analogously to the way previous work used attack success in calibrating $\varepsilon$.

---

[12]https://github.com/privacytrustlab/ml_privacy_meter

### 5.2.2 Results and Discussion

Table 2 presents results for the original chosen set of values of $\varepsilon_G$ from §4, plus two very large values ($\varepsilon_G \in \{200, 300k\}$) to help in understanding and calibrating the behaviour of MIA-R in our context. The table shows that MIA-R is much more successful than the older MIA on our datasets and architectures. For non-private LeNet on CIFAR, the attack AUC is 78.5, and for non-private MLP, it is 82. Success on Fashion MNIST, however, is lower, and close to the chance for LeNet. For the most part, the attacks are sufficiently successful that there is space for calibration of $\varepsilon$.

Looking at the addition of Gaussian noise, we see that the pattern is as we would expect: smaller $\varepsilon$ (larger noise) brings attack success down closer to random chance. In Table 2 we add a couple of larger $\varepsilon$ values beyond those of interest to us for our later comparison of Gaussian and VMF noise, to verify that the pattern continues. For the very tiny Gaussian noise of $\varepsilon_G = 300k$, the AUCs are noticeably larger than for smaller $\varepsilon_G$ but are still well below the non-private attacks for LeNet on CIFAR and MLP on Fashion MNIST.

We also report the train-evaluation gap in Table 2. From it, we observe that the attack success correlates well with overfitting: a large gap means the model achieves a much higher training accuracy compared to unseen samples, which is a characteristic that MIA exploits.

We notice that attacks are more challenging for Fashion-MNIST, and the gaps are usually low. For CIFAR10, gaps below 4% correspond to attacks close to random chance. One needs a gap above 5% to achieve an AUC higher than 55% in both datasets.

AUC under MIA-R thus looks broadly like a good calibration tool for our purposes, although for Fashion MNIST the lower attack success might make calibration more challenging. In addition, the attacks being much less successful for very tiny Gaussian noise than for non-private models (up to 20 percentage points for LeNet on CIFAR10, 58.0 vs 78.5) could be interpreted as suggesting that there is a non-negligible level of protection granted by this noise. We therefore, in §5.3 where we examine data reconstruction as a possible supplementary mechanism for empirical privacy calibration, look particularly at the upper end of the $\varepsilon$ scale.

### 5.3 Data Reconstruction Attacks

We investigate how the noise distributions can defend against attacks during distributed learning, as explained in §2.4. Analogously to MIA, we analyse the impact of a range of $\varepsilon$ on image reconstruction for calibration purposes.

### 5.3.1 Experimental Setup

We consider the DLG attack from Zhu et al. (2019) and the Inverting Gradients method from Geiping et al. (2020). The reasons behind these choices are that (i) DLG is the first reconstruction attack based on gradient sharing, and is well-established as a baseline, and (ii) Inverting Gradients is based on an angular cost function, so we assess whether an angular-based noise like our DIRDP-SGD can defend against it. Next, we explain in more detail each one of these attacks. We set each attack to reconstruct 100 images.

**DLG** An attacker receives the gradients from another participant. Instead of honestly training its neural network, the attacker maliciously uses the gradients to recover the private data that was used to generate them.

Following the notation from Zhu et al. (2019), let $\nabla W$ be the gradients received, $F$ be a twice differentiable neural network, $W$ be its parameters and $(\mathbf{x}, \mathbf{y})$ be the (private) training data and the corresponding (private) labels. The attacker creates dummy $\mathbf{x'}$, $\mathbf{y'}$ (e.g. by sampling from a Gaussian distribution). The dummy data goes through $F$ and after performing backpropagation taking the derivatives w.r.t $W$, the dummy gradients $\nabla W'$ are created:

$$\nabla W' = \frac{\partial \ell(F(\mathbf{x'}, W), \mathbf{y'})}{\partial W} \tag{3}$$

The private training data can be recovered by optimising

$$\mathbf{x'^{*}}, \mathbf{y'^{*}} = \underset{x', y'}{\arg\min} \|W' - W\|^2 \tag{4}$$

More specifically, the attacker takes the difference $\|W' - W\|^2$, which is differentiable w.r.t ($\mathbf{x'}$, $\mathbf{y'}$). Therefore, $\mathbf{x'}$, $\mathbf{y'}$ are optimised by

$$\mathbf{x'}, \mathbf{y'} = \mathbf{x'} - \eta \nabla x' \|W' - W\|^2, \mathbf{y'} - \eta \nabla y' \|W' - W\|^2 \tag{5}$$

where $\eta$ is the learning rate (usually a value in the range $(0, 1]$).

**Inverting gradients (IGA)**   In this attack, Geiping et al. (2020) note that the cost function in Equation 4 optimises a Euclidean matching term, and that the magnitude of a gradient holds information regarding the stage of the training (the gradients tend to be smaller for trained networks). The direction of the gradients can also capture important information, and therefore the authors change Equation 4 to a function based on angles by adopting the cosine distance:

$$\mathbf{x'^{*}}, \mathbf{y'^{*}} = \underset{x \in [0,1]^n}{\arg\min} 1 - \frac{\langle W', W \rangle}{\|W'\|\|W\|} + \alpha TV(x) \tag{6}$$

with the additional constraint that the values in the input data must be normalised to fit the space of $[0, 1]$.

**Metrics**   We consider two metrics that measure how different the reconstructed image is from the original. We attack 100 images.

- **Structural similarity index measure (SSIM)** (Wang et al., 2004) compares any two signals and returns a value between $[-1, 1]$. It compares pixel intensities that have been normalised for luminance and contrast; the work that proposed it demonstrated that it correlates well with human judgements of reconstruction quality. We use it to measure how close the reconstructed images are to the originals; it has previously been used in this way for the specific quantitative evaluation of gradient-based reconstructions (Wei et al., 2020) and (so far non-DP) defences against them (Scheliga et al., 2022), in addition to its longstanding use more generally in image quality assessment (Gu et al., 2020). While there are some complexities in interpreting SSIM scores (Nilsson & Akenine-Möller, 2020), identical images score 1, completely dissimilar images score 0, and negative scores occur rarely and only in unusual contexts.

- **Mean Squared Error (MSE)** measures the distance between a reconstructed image and its original counterpart by averaging the squares of the differences between the pixels of two images. We also use it to measure how similar the reconstructed images are to the original ones. This has likewise been used along with SSIM in the quantitative evaluation of gradient-based reconstruction attacks and their defences.

  As MSE is unbounded, we report its median to avoid a few large MSE values from dominating the average.

We do not consider more recent metrics such as LPIPS (Zhang et al., 2018) that are derived from neural networks and consequently linked to specific architectural choices, e.g. convolutional layers.

### 5.3.2   Results and Discussion

While the DLG attack can reconstruct against the LeNet, it is completely unable to work against the MLP. We note that, although there is nothing in the method that is specific to the target's architecture, Zhu et al. (2019) did only use a CNN as the testbed for experimenting with their attack; perhaps the attack would work against MLPs if modified [13]. However, that is beyond the scope of the present paper. We do note from examining some images (Figure 2) that there is, moreover, little visible distinction between $\varepsilon = 1$ and $\varepsilon = 50$. Overall, this an unsuitable candidate for calibrating between $\varepsilon$s.

---

[13]There's been discussions about the efficacy of this attack when the network is modified, like changing activation functions or even fully training it on some Github issues, see examples `https://github.com/PatrickZH/Improved-Deep-Leakage-from-Gradients/issues/2`, `https://github.com/PatrickZH/Improved-Deep-Leakage-from-Gradients/issues/5`

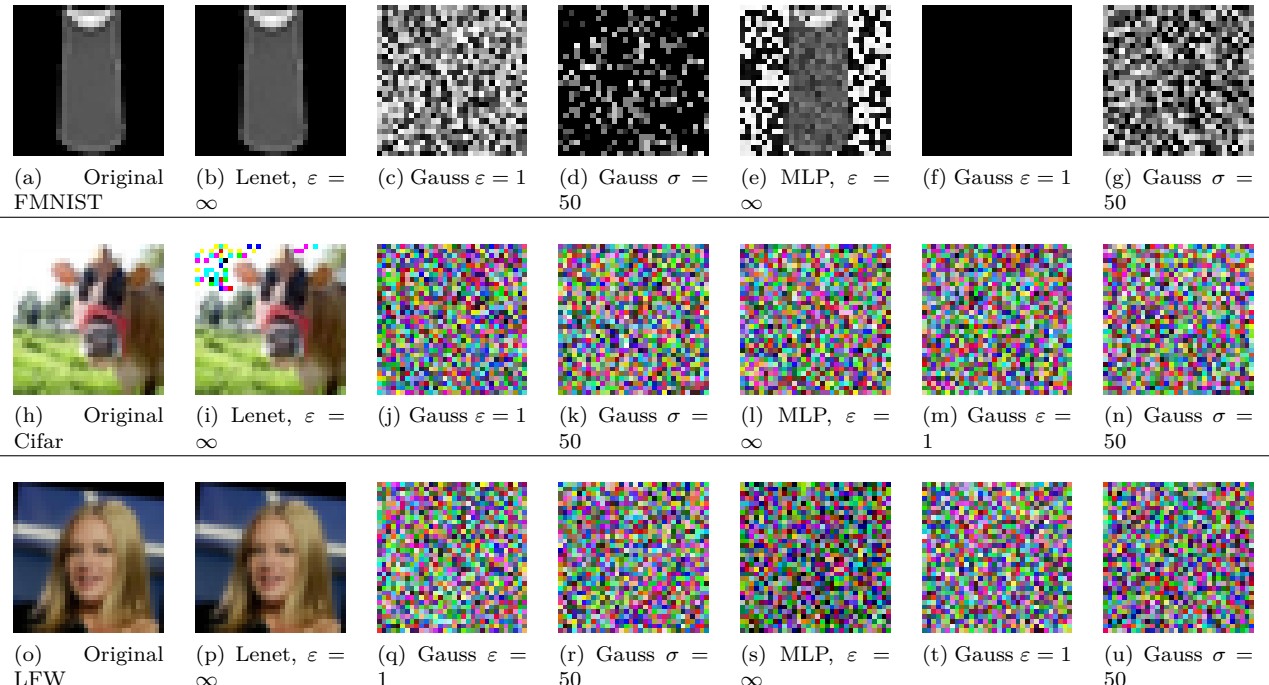

Figure 2: Reconstructed images by DLG against LeNet and MLP.

Table 3: IGA reconstruction attack metrics for LeNet and MLP: Gaussian noise.

| Model | Mechanism | $\varepsilon_G$ | SSIM | | | MSE | | |
|---|---|---|---|---|---|---|---|---|
| | | | **F-MNIST** | **CIFAR** | **LFW** | **F-MNIST** | **CIFAR** | **LFW** |
| LeNet | – | – | 0.29 | 0.28 | 0.24 | 0.65 | 0.36 | 0.68 |
| MLP | – | – | 0.44 | 0.52 | 0.51 | 0.37 | 0.18 | 0.31 |
| LeNet | Gauss | 0.8 | 0.00 | 0.00 | 0.00 | 1.60 | 1.30 | 1.51 |
| | | 1.0 | 0.00 | 0.00 | 0.00 | 1.61 | 1.28 | 1.51 |
| | | 2.0 | 0.00 | 0.00 | 0.00 | 1.60 | 1.25 | 0.49 |
| | | 3.0 | 0.00 | 0.00 | 0.00 | 1.62 | 1.24 | 1.51 |
| | | 8.0 | 0.01 | 0.00 | 0.00 | 1.58 | 1.24 | 1.39 |
| | | 50 | 0.07 | 0.04 | 0.02 | 1.17 | 0.79 | 1.09 |
| | | 200 | 0.23 | 0.14 | 0.05 | 0.71 | 0.48 | 0.96 |
| | | 300k | 0.29 | 0.26 | 0.25 | 0.56 | 0.39 | 0.65 |
| MLP | Gauss | 0.8 | 0.00 | 0.00 | 0.00 | 1.53 | 0.89 | 1.37 |
| | | 1.0 | 0.00 | 0.00 | 0.00 | 1.53 | 0.90 | 1.38 |
| | | 2.0 | 0.00 | 0.00 | 0.00 | 1.54 | 0.88 | 1.37 |
| | | 3.0 | 0.00 | 0.00 | 0.00 | 1.53 | 0.88 | 1.37 |
| | | 8.0 | 0.00 | 0.00 | 0.00 | 1.52 | 0.87 | 1.36 |
| | | 50 | 0.01 | 0.00 | 0.00 | 1.45 | 0.85 | 1.32 |
| | | 200 | 0.03 | 0.02 | 0.01 | 1.36 | 0.78 | 1.26 |
| | | 300k | 0.23 | 0.27 | 0.23 | 0.79 | 0.40 | 0.69 |

Table 3 shows results for the IGA on the same set of values of $\varepsilon_G$ as in §5.2, demonstrating that the attack can succeed against both LeNet and MLP architectures. The values in the table are the average for SSIM and the median for MSE. Higher SSIM values and lower MSE values indicate that the reconstructed image is closer to the original.

We observe that the SSIM scores are all at or close to zero for the originally chosen range of $\varepsilon_G$ ($\varepsilon_G \leq 50$). While, as we have noted, this is a widely used metric for comparing original and distorted images, it is perhaps less suitable here. The kinds of robustness testing this and other metrics undergo involve small amounts of Gaussian and other noise, in addition to affine transformations, JPEG compression, and so on (Gu et al., 2020; Ghildyal & Liu, 2023), but perhaps not in the range of interest in this paper.

The MSE scores, on the other hand, behave broadly as we might wish. As expected, the MSEs for all noisy models are above the non-private ones. Within each architecture, we see the lowest amount of noise at the largest $\varepsilon$. While not too many distinctions can be made among the smaller $\varepsilon$s, it does potentially allow a broad comparison with VMF noise.

We do note at the upper end for the very large $\varepsilon_G$ values as we added in §5.2 to understand the behaviour of MIA-R, that SSIM scores do become positive, and in fact are often greater than the non-private reconstruction SSIM (in one of the six cases for $\varepsilon_G = 200$, and in four for $\varepsilon_G = 300k$). MSE scores start to drop noticeably at $\varepsilon_G = 50$ and thus are somewhat more sensitive. Looking ahead to visualisations in §6 describing the paper's main results, we see that the IGA reconstruction for $\varepsilon_G = 300k$ for our sample images from each of the three datasets can be almost as good as against the non-private models, as indicated by the SSIM scores. This indicates a real privacy leakage not captured by the MIA AUC gap in this kind of empirical privacy calibration. We, therefore, use IGA reconstruction as our second empirical privacy calibration.

### 5.4 Summary

For our empirical privacy calibration, then, we use two tools:

1. the ML Privacy Meter and the success of its MIA-R attacks, as measured by AUC; and

2. the IGA reconstruction attack, focusing on MSE as the metric, as a supplement to the ML Privacy Meter.

## 6 Comparing Gaussian and VMF Mechanisms

In this section, we present results for utility and privacy experiments. More details about hyperparameters and computing environments are described in the Appendix.

### 6.1 Utility experiments

We compare the models after they are trained with and without DP guarantees in classification tasks across different datasets. Table 4 shows the accuracy for each setting for the test sets of Fashion-MNIST, LFW, CIFAR10 and CIFAR100 datasets, after being trained on the respective full training sets.

The first two rows show the test set accuracy of baseline models without any DP mechanism. The remaining rows bring results of the same models, but with different DP mechanisms (Gauss and VMF) and with different values of their respective privacy parameters $\varepsilon_G$ and $\varepsilon_V$.

In general, with few exceptions, adding noise reduces performance, and more noise corresponds to a greater performance reduction: considering $\varepsilon_G$ and $\varepsilon_V$ as privacy budgets, the higher they are, the less privacy should be retained, thus increasing the accuracy.

Overall, in terms of non-private models, relative performance on the datasets and tasks is as expected: Fashion-MNIST is the easiest dataset. Even small values of $\varepsilon$, like $\varepsilon_G = 0.8$ don't seem to drastically affect the utility of the model compared to the non-private baselines. However, we observe other trends when looking at the other datasets.

For the CIFAR dataset, VMF noise leads to much smaller reductions in utility for Lenet and MLP; even the largest $\varepsilon_G = 50$ does not reach the accuracy of the smallest VMF-$\varepsilon_V$. For LFW, there is little difference in performance across the models: the accuracy drop is sharper for Lenet, whereas the bigger $\varepsilon$ approach is closer to the original accuracy. Given its limited size and high number of classes, we report the top-10 accuracy for LFW.

We also observe some cases when the neural network is trained with DIRDP-SGD, its accuracy is even marginally higher than the baseline without any DP guarantees for a large or very large value of $\varepsilon$ (e.g. MLP on Fashion-MNIST, FLW and CIFAR100). We hypothesise that the noise also acts as a regularisation

Table 4: Accuracy scores for classification tasks under different DP settings for the test sets.

| | | | F-MNIST | CIFAR10 | CIFAR100 | LFW |
|---|---|---|---|---|---|---|
| **Model** | **Mechanism** | $\varepsilon_G$, $\varepsilon_V$ | | **Accuracy** | | **Top-10 Acc.** |
| LeNet | – | – | 87.7 | 52.2 | 24.4 | 32.0 |
| MLP | – | – | 84.7 | 45.8 | 16.5 | 15.6 |
| LeNet | Gauss | 0.8 | 81.2 | 36.7 | 4.9 | 8.9 |
| | | 1.0 | 81.7 | 37.4 | 5.5 | 9.6 |
| | | 2.0 | 82.3 | 41.5 | 7.8 | 8.7 |
| | | 3.0 | 83.1 | 43.2 | 9.0 | 8.7 |
| | | 8.0 | 83.9 | 46.6 | 12.0 | 13.2 |
| | | 50.0 | 85.1 | 48.9 | 15.3 | 15.2 |
| LeNet | VMF | 1 | 81.5 | 50.7 | 21.4 | 11.8 |
| | | 5 | 81.3 | 51.3 | 20.7 | 11.2 |
| | | 10 | 81.8 | 50.5 | 21.0 | 11.2 |
| | | 50 | 81.9 | 51.4 | 21.2 | 13.0 |
| | | 500 | 82.9 | 51.4 | 22.2 | 13.9 |
| | | 300k | 83.9 | 51.6 | 25.8 | 14.3 |
| MLP | Gauss | 0.8 | 79.6 | 32.0 | 5.6 | 10.5 |
| | | 1.0 | 79.8 | 32.7 | 5.9 | 10.5 |
| | | 2.0 | 81.2 | 34.6 | 7.0 | 10.3 |
| | | 3.0 | 81.7 | 35.3 | 7.7 | 9.4 |
| | | 8.0 | 82.9 | 36.9 | 8.9 | 9.6 |
| | | 50.0 | 84.1 | 39.1 | 10.6 | 10.7 |
| MLP | VMF | 1 | 84.9 | 42.1 | 13.4 | 14.3 |
| | | 5 | 84.2 | 41.9 | 13.6 | 15.2 |
| | | 10 | 84.4 | 42.3 | 13.8 | 14.7 |
| | | 50 | 84.5 | 42.9 | 14.6 | 15.2 |
| | | 500 | 85.1 | 43.3 | 15.8 | 18.3 |
| | | 300k | 85.3 | 44.8 | 17.0 | 27.1 |

factor that prevents overfitting, which can explain the modest performance gain on the test set. In fact, the use of noise as a regularisation technique has been studied by Li & Liu (2020; 2022), but with Gaussian noise.

## 6.2 Empirical Privacy Calibration

We now present the results of experiments comparing the Gaussian and VMF mechanisms regarding privacy. First, we report the findings on the MIA-**R** attack, followed by the gradient leakage attack.

### 6.2.1 Enhanced MIA

Table 5 gives the results for the enhanced MIA-**R** attack, for both Gaussian and VMF noise. We left the LFW dataset out of this set of experiments due to its limited size to be split into several training partitions for the reference models.

The calibration with respect to standard Gaussian noise of §5.2 showed that the attack was particularly successful against CIFAR, with a large improvement over chance against the non-private models. Comparing Gaussian against VMF on LeNet, we see that even the largest $\varepsilon_V$ reduces the success of the attack to less than the Gaussian $\varepsilon = 1.0$.

For Fashion-MNIST, the baseline models without any privacy guarantee leak less information than those trained with CIFAR. Thus, unsurprisingly, most noise settings make the attacks similar to random chance, except for some cases where $\varepsilon$ is very large, particularly with MLP.

Table 5: AUC attack scores under different DP settings for the Enhanced Membership Inference Attacks using the ML Privacy Meter (MIA-R), comparing Gaussian and VMF noise.

| Model | Mechanism | $\varepsilon_G$, $\varepsilon_V$ | F-MNIST Attack AUC | CIFAR10 Attack AUC |
|---|---|---|---|---|
| LeNet | – | – | 54.8 | 78.5 |
| MLP | – | – | 61.1 | 82.0 |
| LeNet | Gauss | 0.8 | 49.2 | 50.9 |
| | | 1.0 | 49.3 | 51.1 |
| | | 2.0 | 50.1 | 51.3 |
| | | 3.0 | 49.9 | 51.2 |
| | | 8.0 | 50.5 | 51.6 |
| | | 50.0 | 50.4 | 51.7 |
| LeNet | VMF | 1.0 | 49.5 | 50.8 |
| | | 5.0 | 48.9 | 50.5 |
| | | 10.0 | 49.3 | 50.4 |
| | | 50.0 | 49.8 | 50.8 |
| | | 500.0 | 50.4 | 57.1 |
| | | 300k | 51.8 | 66.0 |
| MLP | Gauss | 0.8 | 49.8 | 50.5 |
| | | 1.0 | 49.9 | 50.5 |
| | | 2.0 | 50.4 | 51.0 |
| | | 3.0 | 51.1 | 51.4 |
| | | 8.0 | 51.8 | 52.0 |
| | | 50.0 | 52.6 | 53.9 |
| MLP | VMF | 1.0 | 49.2 | 50.3 |
| | | 5.0 | 48.7 | 50.2 |
| | | 10.0 | 49.2 | 50.6 |
| | | 50.0 | 49.7 | 51.9 |
| | | 500.0 | 52.3 | 58.9 |
| | | 300k | 55.1 | 74.3 |

### 6.2.2 Defence against Gradient Leakage Attacks

As a supplement to the Enhanced MIA calibration, Table 6 shows the results of the IGA (Inversion Gradient Attack) reconstruction attack against LeNet and MLP models, measuring the ability of an attacker to reconstruct the input data from the model's output probability distribution.

The MSE metric can be seen as the amount of noise that was kept in the resulting image after the attack was completed. We observe that reconstruction against LeNet generally performs worse than the MLP model. For instance, in terms of the SSIM metric, the LeNet model has a maximum score of 0.29, while the MLP model has a maximum score of 0.52. Also, the LeNet model has a higher MSE score than the MLP model for all datasets.

When we look at the performance of the models under attack, we can see that for the most part, the reconstructions under Gaussian and VMF DP are noisier than the non-private reconstructions. The exception is, as expected, the very large $\varepsilon_V = 300k$ that we use for understanding the range of $\varepsilon_V$s, where the MSE scores are similar to or only a little larger than for the non-private models. This is also reflected in the SSIM scores, where with a VMF mechanism and an $\varepsilon_V = 300k$, the LeNet model has an SSIM score of 0.28 for the F-MNIST dataset, and the MLP model has an SSIM score of 0.27 for the same attack configuration. This indicates that both LeNet and MLP models can be vulnerable to the IGA attack if the noise is too small.

We also see that when Lenet's gradients receive Gaussian noise, the error drops sharply as $\varepsilon$ increases, mainly for Fashion-MNIST and CIFAR images. For the remaining experiments, the range of values is much

Table 6: IGA reconstruction attack metrics for LeNet and MLP.

| Model | Mechanism | $\varepsilon_G/\varepsilon_V$ | SSIM | | | MSE | | |
|---|---|---|---|---|---|---|---|---|
| | | | F-MNIST | CIFAR | LFW | F-MNIST | CIFAR | LFW |
| LeNet | – | – | 0.29 | 0.28 | 0.24 | 0.65 | 0.36 | 0.68 |
| MLP | – | – | 0.44 | 0.52 | 0.51 | 0.37 | 0.18 | 0.31 |
| LeNet | Gauss | 0.8 | 0.00 | 0.00 | 0.00 | 1.60 | 1.30 | 1.51 |
| | | 1.0 | 0.00 | 0.00 | 0.00 | 1.61 | 1.28 | 1.51 |
| | | 2.0 | 0.00 | 0.00 | 0.00 | 1.60 | 1.25 | 0.49 |
| | | 3.0 | 0.00 | 0.00 | 0.00 | 1.62 | 1.24 | 1.51 |
| | | 8.0 | 0.01 | 0.00 | 0.00 | 1.58 | 1.24 | 1.39 |
| | | 50 | 0.07 | 0.04 | 0.02 | 1.17 | 0.79 | 1.09 |
| LeNet | VMF | 1 | 0.00 | 0.00 | 0.00 | 1.80 | 1.46 | 1.51 |
| | | 5 | 0.00 | 0.00 | 0.00 | 1.79 | 1.48 | 1.56 |
| | | 10 | 0.01 | 0.00 | 0.00 | 1.67 | 1.45 | 1.53 |
| | | 50 | 0.01 | 0.00 | 0.01 | 1.66 | 1.39 | 1.29 |
| | | 500 | 0.10 | 0.06 | 0.02 | 1.05 | 0.77 | 1.07 |
| | | 300k | 0.28 | 0.27 | 0.16 | 0.58 | 0.37 | 0.75 |
| MLP | Gauss | 0.8 | 0.00 | 0.00 | 0.00 | 1.53 | 0.89 | 1.37 |
| | | 1.0 | 0.00 | 0.00 | 0.00 | 1.53 | 0.90 | 1.38 |
| | | 2.0 | 0.00 | 0.00 | 0.00 | 1.54 | 0.88 | 1.37 |
| | | 3.0 | 0.00 | 0.00 | 0.00 | 1.53 | 0.88 | 1.37 |
| | | 8.0 | 0.00 | 0.00 | 0.00 | 1.52 | 0.87 | 1.36 |
| | | 50 | 0.01 | 0.00 | 0.00 | 1.45 | 0.85 | 1.32 |
| MLP | VMF | 1 | 0.00 | 0.00 | 0.00 | 1.54 | 0.91 | 1.38 |
| | | 5 | 0.00 | 0.00 | 0.00 | 1.55 | 0.91 | 1.39 |
| | | 10 | 0.00 | 0.00 | 0.00 | 1.52 | 0.90 | 1.36 |
| | | 50 | 0.00 | 0.00 | 0.00 | 1.52 | 0.89 | 1.38 |
| | | 500 | 0.00 | 0.00 | 0.00 | 1.47 | 0.86 | 1.33 |
| | | 300k | 0.17 | 0.18 | 0.12 | 0.91 | 0.50 | 0.86 |

lower, even though the same trend can be observed in some cases, as in the MLP with Gaussian noise for Fashion-MNIST.

Overall, though, the MSE values when comparing like setups for Gaussian and VMF are similar (e.g. for CIFAR MLP, both are around 0.9 for their ranges, with the exception of $\varepsilon_V = 300k$ as noted above). Where this is not the case, it is broadly in favour of VMF: for example, for the smaller $\varepsilon$ for LeNet on Fashion MNIST and CIFAR, the noise added by VMF is greater than Gaussian. This then provides less fine-grained calibration than the Enhanced MIA but does broadly support it, and provides an additional anchor point for the empirical privacy calibration.

We give some examples of IGA reconstructions in Figure 3, for very small and very large $\varepsilon$. It can be seen that only for the exceptional $\varepsilon_V = 300k$ is any reconstruction really visible, supporting the interpretation of the metrics.

### 6.2.3 Calibration Outcome

For each of the datasets, both methods of calibration indicate that the nominal values of $\varepsilon_G$ and $\varepsilon_V$ are comparable for ranges $0.8 \leq \varepsilon_G \leq 50$ and $1 \leq \varepsilon_V \leq 50$, in those cases where VMF is not better. To facilitate rapid comparison for those $\varepsilon$s where the nominal values are the same $(1, 50)$, in Table 7 we present the differences between attack success rates under the MIA-R attack, and the MSE scores under the IGA attack. In the former case, negative scores are better for VMF; in the latter, positive scores are better for VMF. It is only for the LeNet on Fashion-MNIST for $\varepsilon = 1$ under MIA-R where Gaussian is better, and in this case, the MIA is only at chance, so there is essentially no difference between the two.

Table 7: Comparison of empirical privacy. $\Delta_{\text{MIA-R}}$ indicates MIA-R attack success under Gaussian noise minus attack success under VMF noise; negative is better for VMF. $\Delta_{\text{IGA}}$ indicates IGA MSE under Gaussian noise minus MSE under VMF noise; positive is better for VMF.

| Model | $\varepsilon_G/\varepsilon_V$ | $\Delta_{\text{MIA-R}}(\downarrow)$ | | $\Delta_{\text{IGA}}(\uparrow)$ | | |
| | | F-MNIST | CIFAR | F-MNIST | CIFAR | LFW |
|---|---|---|---|---|---|---|
| LeNet | 1 | 0.2 | -0.3 | 0.19 | 0.18 | 0 |
| | 50 | -0.6 | -0.9 | 0.49 | 0.6 | 0.2 |
| MLP | 1 | -0.7 | -0.2 | 0.01 | 0.01 | 0 |
| | 50 | -2.9 | -2 | 0.07 | 0.04 | 0.06 |

Having established that the nominal $\varepsilon$ values are broadly comparable, we can verify looking back at Table 4 that the VMF attack is almost always better for similar levels of Gaussian and VMF noise.

### 6.3  Note on Efficiency

Overall we found that VMF noise is able to protect against gradient-based reconstruction attacks and to offer good levels of utility in image classification datasets over different datasets. This is particularly illustrated in the LFW dataset, which is based on face images where one might wish to hide their identity.

However, VMF can be computationally expensive for gradients with high dimensionality, and future work involves optimising it. This is in line with previous studies that enhanced vanilla approaches for DP in deep learning, like backpropagation improvements such as ghost clipping from Li et al. (2022a), and the efficient per-sample gradient computation from Yousefpour et al. (2021).

## 7  Conclusions

We defined DIRDP-SGD for directional privacy in deep learning by adding noise to the gradients during training. This problem is particularly relevant because several studies have shown that private training data can be discovered under certain machine learning training settings, such as sharing gradients.

Our mechanism provides both $\varepsilon d$-privacy and $\varepsilon$-DP guarantees rather then $(\varepsilon, \delta)$-DP. Because the $\varepsilon$s are not analytically comparable across frameworks, we analyse both membership inference attacks (MIAs) and gradient-based reconstruction attacks as possible methods for calibrating privacy leakage. We show that the enhanced MIA of Ye et al. (2022) and the Inverted Gradient Attack of Geiping et al. (2020) are useful for calibrating nominal values of $\varepsilon$ across standard DP and metric DP frameworks. Given this empirical privacy calibration framework, our experiments showed that DIRDP-SGD can provide better utility than standard DP for similar levels of privacy. Future work that extends beyond this initial mechanism proposal would look at other kinds of empirical privacy audits, for example, the recently proposed generative gradient leakage of Li et al. (2022b).

DIRDP-SGD is based on the VMF distribution and can be computationally expensive for high-dimension data. Future work here would include optimising the mechanism in the same way that DP-SGD has seen much effort in improvements, such as in the efficient processing of gradients or choice of activation functions. Moreover, experiments were restricted to image datasets. We plan to explore the feasibility of DIRDP-SGD for other domains, such as natural language processing.

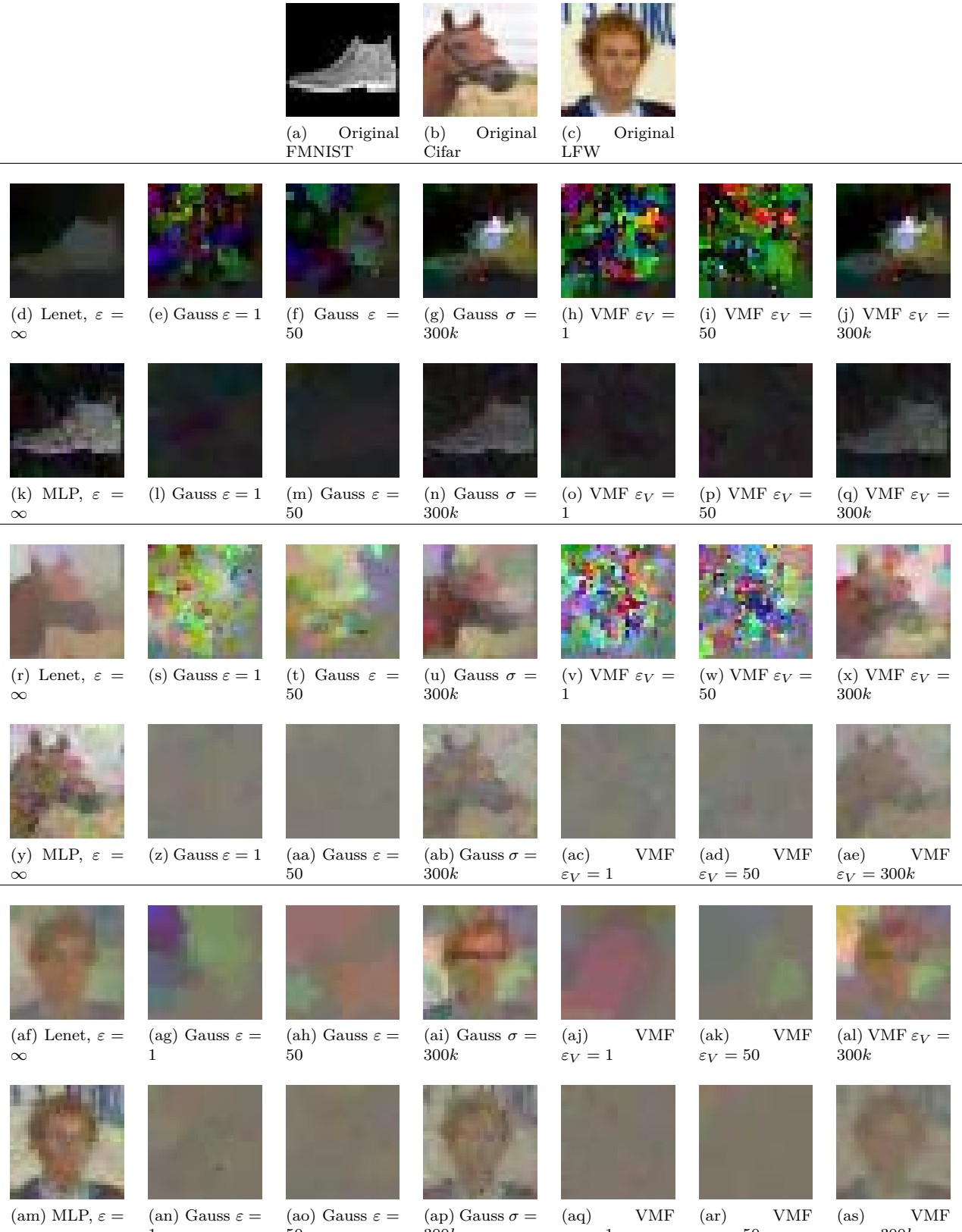

Figure 3: Reconstructed images by Inverting Gradients against LeNet and MLP.

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
