# OpenReview forum: "Directional Privacy for Deep Learning"
_TMLR — Rejected by TMLR_

### Review · Reviewer_pu4X · 2023-07-16

**Summary Of Contributions:**

The paper proposes a new DP-SGD algorithm based on the von Mises-Fisher (VMF) distribution to achieve directional privacy. Their experiments on key datasets indicate that the VMF mechanism can outperform the Gaussian mechanism in the utility-privacy trade-off.

**Audience:**

Yes

**Broader Impact Concerns:**

This work does not describe any potential negative social impact of the results presented.

**Claims And Evidence:**

Yes

**Requested Changes:**

1. Algorithm 2 should be described more clearly. Specifically, in line 13, what is the explicit expression of $\mathcal{V}(\sigma, \bar{g}_t(x_i))$?

2. In Definition 3, there is a factor $C_K(\epsilon)$.  Will it affect the outcome of the experiment? Even if it is a constant, it cannot be ignored.

3. The proof needs to be more rigorous and self-consistent.

**Strengths And Weaknesses:**

Strength

Experiments show the effectiveness of the proposed algorithm.

Weakness.

1. In general, the paper is worse written. Specifically, many symbols are not defined. For example, on Page 6, What does “for all $x\sim x’ \in \mathcal{X}$” mean? The definition of the neighboring datasets is not defined. On page 8, norm $||\cdot||$ is not defined. In addition, some definitions lack discussions and clarifications. For instance, the definition of metric DP (Def 1) needs some discussion (examples or relationship between it and pure DP); Is directional privacy a special case of metric DP? Also, the organization of the paper is not good, I would suggest the authors move section 3.1 to the experiment section.


2. The statements and proofs of theorems are not rigorous and not self-consistent (especially the proof of Theorem 2).  It’s unclear to me why Theorem 2 holds. And why do the authors consider the privacy with respect to batches instead of the whole algorithm (the privacy loss of the total $T$ iterations)? It seems that the latter is common in the literature. In addition, the post-processing property is not introduced.

---

> ### Author Response · Authors · 2023-07-26
> **Addressing Review of Paper1282 by Reviewer pu4X**
>
> We thank the reviewer for providing valuable feedback and directions. We address their issues below.
>
> Weakness 1:
>     We thank the reviewer for highlighting clarifications we need to make. Directional privacy is an instance of metric DP, where the metric is one based on angular distance of vectors. We will clarify this further in the paper. We will be happy to add an example of metric DP drawn from the literature noted in Sec 2.1 to illustrate the relationship with pure DP. We have broadly followed the structure of previous papers like the classic Abadi et al (2016) [3], where all the formal privacy material, including definitions of DP-SGD, are presented in a separate section before the experimental setup. Some definitions, like neighbouring datasets, are inherent to differential privacy since its proposal in 2006; metric DP generalises this (so that the notion of neighbouring datasets is the special case of the metric being the Hamming distance, as we note in Sec 1)..
>
> Weakness 2:
>     We have followed the proposal of DP-SGD by Abadi et al [3]. Their paper (see e.g. the lead-up to Thm 1 of [3]) analyses results w.r.t. batches (or in terms of 'lots', as per their terminology). Post-processing happens in every part of the training where gradients are used after noise has been added. More specifically, in the context of DP-SGD, the parameter updates done by the optimisers are the post-processing steps. The post-processing property of differential privacy (aka data processing inequality for DP) was established by Dwork et al [5, Proposition 2.1, p19]. It was generalised to d-privacy by Chatzikokolakis et al. [6] and also used in [7]. We can include explicit reference to post-processing and the data processing inequality in our revised paper.
>
> Requested changes:
>
> 1) V stands for sampling from a von Mises-Fisher distribution, sigma controls the amount of noise and gt(xi) refers to the sample gradients. Our notation is consistent with the notation of Algorithm 1, which samples from the Gaussian distribution and was used in [3] when DP-SGD was first proposed, and also with the notation used in the cited Weggenmann and Kerschbaum (2021) paper that introduced directional privacy.
>
> 2)  This factor is just a normalisation constant that ensures that the full expression is a valid density function; there is an analogous one for a multivariate Gaussian. The dimension K refers to the dimensionality of the model’s layers, so it’s not something that we tune. Just as with the Gaussian, it has no effect on the outcome of experiments.
>
> 3)  We will clarify the explanation about the steps taken by the mechanism by adding in more details from the citations to the literature that are used to justify various proof steps.
>
>
> Broader Impact Concerns:
>     The use of a directional privacy mechanism in SGD does not introduce any new potential negative social impact beyond the use of DP in training machine learners. In general, privacy is seen as potentially contributing positively. There is some discussion in the DP literature about unintended possible negative disparate impacts (i.e. that different groups might be affected differently), although there is no agreement about this [2]. Also, our approach aligns with [1], which states that many DP-based ML implementations basically add noise similar to other critcised statistical disclosure control works, and the actual level of privacy must be experimentally evaluated, which is precisely what we assess.
>
>
> [1] A Critical Review on the Use (and Misuse) of Differential Privacy in Machine Learning, https://dl.acm.org/doi/10.1145/3547139
>
> [2] Disparate Impact in Differential Privacy from Gradient Misalignment, https://openreview.net/forum?id=qLOaeRvteqbx
>
> [3] Deep Learning with Differential Privacy, https://dl.acm.org/doi/10.1145/2976749.2978318
>
> [4] (Local) Differential Privacy has NO disparate impact on fairness, https://arxiv.org/pdf/2304.12845
>
> [5] The Algorithmic Foundations of Differential Privacy, Dwork et al, https://dl.acm.org/doi/10.1561/0400000042
>
> [6] Broadening the Scope of Differential Privacy using Metrics, Chatzikokolakis et al, https://inria.hal.science/hal-00767210/document
>
> [7] Universal optimality and robust utility bounds for metric differential privacy, Fernandes et al. 2022.

---

### Review · Reviewer_1Bkc · 2023-07-28

**Summary Of Contributions:**

This paper proposes DirDP-SGD a variant of DP-SGD that satisfies directional differential privacy. An empirical framework is proposed for comparing DP algorithms under different flavours of DP, based on membership inference and data reconstruction attacks. This framework is then used to compare DirDP-SGD with the usual variant of DP-SGD in training of DNNs on image classification tasks, where it can yield better results for comparable privacy guarantees.

**Audience:**

Yes

**Broader Impact Concerns:**

I am slightly worried that the proposed empirical evaluation framework of DP may be biased; I would either add a dedicated paragraph about this near Section 6.2 (or in the conclusion), or strengthen the empirical comparison by analyzing the empirical behavior of more methods/choice of hyperparameters.

Nonetheless, the study of this type of algorithms is a very interesting direction for the DP community, and could have a large, positive, impact (i.e. helping in obtaining better privacy/utility trade-offs).

**Claims And Evidence:**

No

**Requested Changes:**

1. *(Critical for Acceptance.)* Mathematical proofs should be made more clear. In particular, Corollary 1's proof is a bit evasive, and privacy guarantees from Corollaries 2 and 3 seem to ignore the fact that the algorithm is iterative. It also seems that under the standard composition/amplification setting, DirDP would satisfy approximate DP just like DP-SGD. Alternatively, DirDP-SGD could also be compared with a pure DP-SGD algorithm based on the Laplace mechanism.
2. *(Critical for Acceptance.)* The calibration procedure for the MIA/Reconstruction attacks seem a little bit light. I would appreciate either developing discussion on why this is reasonable, or maybe showcase the efficiency of such methods on other methods (like output perturbation) or different set of hyperparameters, which would help to ensure that calibration is fair for all proposed methods.
3. *(Less Critical.)* It would be nice to have empirical discussions on how the difference between DP-SGD and DirDP-SGD evolves with the parameters of the problem; ideally, this could highlight settings where DirDP-SGD can exploit structural properties of the data that DP-SGD cannot.
4. *(Less Critical.)* The organization of the paper makes it difficult to follow. Contributions are a bit hidden in pages of experimental details, that could be put in appendix. I would also suggest re-organizing Section 5 so that contributions are more clear, and less time is spent on the description of existing methods (that could be described in related work or appendix).
5. *(Not Critical/Would Strengthen the Work.)* Providing theoretical guarantees on DirDP-SGD's utility would be a very nice contribution (even if only in convex settings); in particular if it allows discussing dependence on the dimension more clearly.


**Strengths And Weaknesses:**

**Strengths**
1. The paper proposes an interesting and very promising variant of DP-SGD (the most commonly used algorithm in DP optimization) under directional differential privacy, which can achieve a better privacy-utility trade-off on some problems.
2. When gradients live in a lower-dimensional subspace, DirDP-SGD seems to be able to improve dependence on the dimension (although this is not formally proven).
3. A general framework, based on membership inference and reconstruction attacks, is proposed to compare DP optimization algorithms that satisfy different variants of DP.

**Weaknesses**
1. Mathematical proofs are a bit evasive:
   - In Corollary 1: I did not understand what is the "standard property" used to derive the inequality on $\mathcal{V}^*$, and where does the $\sqrt{m}$ come from.
   - Corollaries 2 and 3 gives a guarantee for Algorithm 2 assuming batches are disjoint, but there is no reason this is the case (at least in the way Algorithm 2 is stated). Such assumption is not standard in usual analyses of DP-SGD (which rely on DP composition and amplification results).
2. Although DP-SGD satisfies approximate DP (and not pure DP), it seems that it would also be the case for DirDP-SGD with advanced composition (or RDP composition). It is not clear why comparing the two algorithm is not reasonable. Conversely, DP-SGD could be made pure DP with Laplace noise and basic composition results of DP.
3. The MIA part of the empirical evaluation framework is only studied on DP-SGD for an arbitrary set of hyperparameters. Additionally, in Tables 2 and 3, significant changes are only observed when $\epsilon \ge 50$, which gives very limited privacy guarantees. Thus, it is not clear whether the proposed methodology properly assesses privacy guarantees on all algorithms in higher privacy regimes (e.g., when $\epsilon < 1$). It also sounds somewhat biased to compare the VMF mechanism (which protects angular distance) with the Gaussian mechanism using gradient inversion attacks based on angular distance.
4. An important problem in DP optimization is that high-dimensional models require the addition of important amounts of noise. While the discussion after Corollary 1 seems to go towards this direction (reducing dependence on dimension from $n$ to $m$), hinting that the proposed method could reduce dependence on dimension in some settings, this is not studied at all in the rest of the paper.

Although this is not necessarily a problem since the paper is quite empirical, no theoretical guarantees are given on the utility of DirDP-SGD. It seems that there exists convergence results on DP normalized SGD (see e.g., [1]), these results could maybe be extended to DirDP-SGD.

There are some typos in the manuscript. For instance:
- in $d_\theta (v, v')$, the argument of the arccos should be normalized (there the value of arccos itself is normalized, which seems a bit odd).
- in Theorem 2, $\mathcal{V}M$ refers to VMF?
- the value of $\sigma$ should be stated in the statement of Corollaries 2 and 3.

Finally, [2] may be an interesting reference to discuss alongside Rezaei & Liu (2021).

**References:**

[1] Normalized/Clipped SGD with Perturbation for Differentially Private Non-Convex Optimization, Yang et al., 2022.

[2] Membership Inference Attacks From First Principles, Carlini et al., 2021.

---

> ### Author Response · Authors · 2023-08-11
> **Response to Reviewer 1Bkc**
>
> We thank the reviewer for the feedback and directions
>
> About Corollary 1: We can provide a full proof of the mathematical results. The standard property of DP refers to the privacy guarantee for independent applications of a perturber (VMF) to the components of a vector (expressed as a sum of orthonormal vectors). The proof of this is in [Dwork&Roth 2014], but we can make this clearer. The “m” variable in the corollary refers to the way a point can be represented as convex sum of m orthogonal vectors in n-dimensions: this gives a better sqrt(m) factor rather than sqrt(n); this stronger bound is possible with VMF.
>
> Regarding Corollaries 2 and 3: As the first work on applying metric DP to SGD, we followed the first approach (Song et al (2013)) to apply pure eps within standard DP; we generalised that original proof, which did assume that the batches are disjoint. This is also consistent with the later Abadi et al (2016), where construction of batches is done by random permutation followed by partitioning, giving disjoint sets (Sec 3.1).  We’ll explicitly note this in the paper.
>
> Regarding comparing epsilons across frameworks in a theoretical way, this kind of theoretical comparison is interesting for future work to allow more direct comparisons. However, converting the VMF noise via RDP to extract the usual privacy parameters I.e. eps and delta for direct comparison is not straightforward, and is beyond the scope of our paper. It is true that one can turn DP-SGD into pure-DP with Laplace, but the de facto standard for ML has been the Gaussian mechanism.
>
> To respond to the remark that using gradient inversion is potentially biased towards VMF, our principal reason for choosing this attack is that it and the earlier DLG are well-established attacks with available code, but as we note, DLG was not successful in any reconstructions under our MLP model.  We do observe that DP-SGD itself is also aiming to explicitly perturb the direction / angle of the gradient as the basis of the privacy mechanism (e.g. discussion in Sec 3.1 of Abadi et al, 2016), but we recognise that the VMF mechanism is more directly linked to the angle as a metric.
>
> Regarding the suggestion concerning high dimensionality and reducing dependence on dimension from n to m, this is indeed an interesting direction for future research. Our goal is to first establish the use of VMF. Thus, optimisations fall out of the scope of our current paper.
>
> Regarding the requested changes:
>
> For those under Critical for Acceptance (mathematical proofs), most of our response is given above.  Essentially, we will add some further detail.  Regarding the iterative aspect, it is true that our results are a generalisation of Song’s work for a single iteration of the algorithm. The use of the accountant in Abadi et al’s work provides an opportunity to provide a tighter bound for an overall epsilon estimate by making use of an increase in delta. For VMF we can certainly use the DP composition results; however a better and more reasonable comparison to Abadi et al’s approach would be to use RDP for the iterative part, but as mentioned above this is out of scope for this study and we leave it to future work.
>
> Regarding those under Critical Acceptance (empirical privacy), again, much of our response is to be found above.  We’d note that what we have already is an advance beyond previous literature which only used (original) MIA, which we showed not to be helpful in our context.  As we mentioned above (and also in response to Broader Impact Concerns), we can add some further discussion, particularly about the reconstruction attack, plus some further results on more hyperparameters (epsilon, MIA reference models).
>
> About the Less Critical (empirical differences between DP-SGD, DirDP-SGD), this would be good, although we haven’t discovered any patterns where we have an interesting explanation for our observations.  (For example, why is FMNIST so different from the other two, in terms of the comparative utilities under DP-SGD vs DirDP-SGD? The greater relative simplicity of the task? The type of image? We intend to look at this in future work.
>
> For those under Less Critical (organisation): This is a good suggestion we'll follow
>
> For those under Not Critical: this is a very interesting direction that we have in fact already been looking into.  It does seem to have a number of complexities that were not apparent when we first started looking into it, so it is very much future work.
>
> [3] Differentially Private Learning with Grouped Gradient Clipping, https://dl.acm.org/doi/fullHtml/10.1145/3469877.3490594
>
> [4] Privacy-Preserving Graph Convolutional Networks for Text Classification, https://aclanthology.org/2022.lrec-1.36/
>
> [5] Building an Image Classifier with Differential Privacy, https://opacus.ai/tutorials/building_image_classifier
>
> [6] DP-BART for Privatized Text Rewriting under Local Differential Privacy, https://aclanthology.org/2023.findings-acl.874.pdf

---

> > ### Comment · Reviewer_1Bkc · 2023-08-13
> >
> > Thank you very much for your response, that clarifies most of my questions.
> >
> > Regarding my third question on DP-SGD vs. DirDP-SGD, I was in fact wondering whether one can create a toy problem/dataset where DirDP-SGD is expected to perform better (e.g. a task that heavily and explicitly relies on angles between records, rather than on something else), and where it manages to get arbitrarily good improvement over DP-SGD numerically.

---

> > > ### Author Response · Authors · 2023-08-17
> > >
> > > We believe this would be an excellent addition. Unfortunately, we don't have such a scenario where DirDP-SGD is guaranteed to have an arbitrarily good improvement over DP-SGD (although we’ll keep trying to think of one).  However, we have constructed a toy example that can illustrate how VMF is working more effectively. The example is based on a minimal MLP for XOR, including only six weights (no bias, no activation functions), and we can show that with a similar (or lesser) amount of gradient perturbation as measured by cosine similarity or Euclidean distance, VMF noise leads to better utility.  We have a figure illustrating this -- with all the weights and their associated gradients, alongside the numerical output, making explicit the deviations for the weights according to each set of gradients – and can include this in the paper.

---

### Review · Reviewer_9Q3m · 2023-08-06

**Summary Of Contributions:**

This paper applies the von Mises-Fisher (VMF) mechanism to stochastic gradient descent. The paper calls the resulting SGD method DirDP-SGD because the VMF mechanism generally can guarantee directional differential privacy (DirDP).

In order to compare the performance of the DirDP-SGD with DP-SGD (which is based on the Gaussian mechanism), the paper considers membership inversion attacks and gradient reconstruction attacks.

**Audience:**

Yes

**Claims And Evidence:**

Yes

**Requested Changes:**


The empirical study needs to be significantly modified. In its current form, it does support the authors' claims. Please see the weaknesses in the previous section.

The authors also should clarify their contributions. I understand that they are applying VMF to SGD. Is there any more contributions than applying VMF to SGD?

**Strengths And Weaknesses:**

Strength:

1. Applying VMF to SGD and providing empirical study
2. Providing privacy analysis for DirDP-SGD

Weaknesses:

1. The empirical study has a lot of redundant tables. For example, table 1 and Table 2 provide results for membership inversion attacks only for the Gaussian mechanism. However, the goal of empirical study is to compare the Gaussian mechanism with VMF.

2. The tables or figure does not provide any variance/standard deviation for accuracy/AUC or other metrics. This makes it difficult to compare different methods.

3. The numerical results provided in the tables make the comparison between DP-SGD and DirDP-SGD very difficult. DP-SGD and DirDP-SGD are two different mechanisms and their privacy parameters cannot be directly compared. This is because DP-SGD provides \epsilon,delta differential privacy and DirDP-SGD guarantee directional DP. As a result, the authors suggest using attack probability (or MSE for gradient reconstruction) as a privacy measure. Under the same attack probability, the mechanism that has better accuracy has a better privacy-accuracy trade-off. However, the authors provide accuracy and attack probability in two separate tables. Why we cannot have them in a single table. For a fixed attack probability, we should compare the accuracy of the two methods. I also suggest that the authors provide a couple of figures. In the figure, the x-axis should be the attack probability (or MSE for gradient reconstruction attack) and y-axis should be the accuracy. Given this figure, we can compare the privacy-accuracy trade-off. Not with the tables provided in the paper.

For the tables, also the authors should provide the attack probability in one column and the accuracy of DP-SGD in another column, and the accuracy of DirDP-SGD in another column.

---

> ### Author Response · Authors · 2023-08-11
> **Response**
>
>
>
> In terms of the comment about redundant tables: We do compare the Gaussian mechanism with VMF.  As we explain in Sec 5, we first examine the MIA approach from previous literature, which was there applied to the Gaussian mechanism under particular experimental setups, to see whether that carries across to our Gaussian setup.  We show that under the previously used MIA it does not – this is the purpose of Tables 1 and 2 (Table 1 that the vanilla MIA performs poorly. Table 2 shows better attacks from MIA-R) – and so we examine instead the more recent MIA-R, which we show does work in the intended way for the Gaussian mechanism.  We can then use it to compare Gaussian and VMF; that’s why we compare VMF against the Gaussian in Table 5 for MIA-R only.
>
>
>
> In terms of calculation of standard deviations, we note that while these are desirable, none of the previous work that we are following on from report variance or standard deviation of any experiment [3, 4, 5, 6, 7].  This is no doubt in part because the experiments are computationally expensive, as we have remarked in Sec 6.3.
>
>
>
> Regarding a single presentation combining utility and privacy, it’s not clear what a single table would look for that.  “For a fixed attack probability, we should compare the accuracy of the two methods”: while we can fix epsilon, we can’t fix attack success.  However, the suggestion of plots is a good one.  We have experimented with scatter plots that show the trade-off between privacy (one for MSE and another for AUC on the x-axis) and utility (accuracy on the y-axis), in a way that graphically shows what we have described in words in Sec 6.2.
>
>
>
> Requested changes #1: This is in our response to weaknesses above.
>
>
>
> Requested changes #2: Our contributions are as we have listed in Sec 1:
>
> * We apply for the first time a metric DP mechanism based on angular distance --- via the von Mises-Fisher distribution --- to use as an alternative to Gaussian noise in training via Stochastic Gradient Descent in deep learning;
>
> * We demonstrate that this provides $\epsilon d_\theta$-privacy (for angular distance $d_\theta$) as well as $\epsilon$-DP  for the training as a whole;
>
>  * We analyse both MIAs and gradient-based reconstruction attacks as candidates for empirically comparing privacy, and show why using both together is appropriate in this context.
>
>
>
>
>
> * Given this, we show that overall on major datasets, our VMF mechanism outperforms Gaussian noise when defending against attacks.
>
> * We find that the $\epsilon$ parameter used in both mechanisms is in general \emph{not informative} for comparing mechanisms with respect to their defence against the reconstruction attacks we used. In particular, we find that the von Mises Fisher mechanism can defend against these attacks even using very high values for $\epsilon$ while providing good utility.
>
>
>
> So, for example, we see our contribution to applying VMF to SGD as including distinct theoretical and experimental contributions; another contribution in the list is that we go beyond the previous literature in our empirical privacy evaluation (verifying that the previous MIA does not work in our context, applying MIA-R, and adding the reconstruction attack).
>
>
>
>
>
> [1] Differential privacy for location-based systems. https://dl.acm.org/doi/10.1145/2508859.2516735
>
> [2] Privacy- and Utility-Preserving Textual Analysis via Calibrated Multivariate Perturbations.   https://dl.acm.org/doi/10.1145/3336191.3371856
>
> [3] Deep Learning with Differential Privacy, https://dl.acm.org/doi/10.1145/2976749.2978318
>
> [4] Removing Disparate Impact on Model Accuracy in Differentially Private Stochastic Gradient Descent https://dl.acm.org/doi/10.1145/3447548.3467268
>
> [5] Differentially Private Learning with Grouped Gradient Clipping https://dl.acm.org/doi/10.1145/3469877.3490594
>
> [6] Large Language Models Can Be Strong Differentially Private Learners https://openreview.net/forum?id=bVuP3ltATMz
>
> [7] Differentially Private Learning Needs Better Features (or Much More Data). https://openreview.net/pdf?id=YTWGvpFOQD-

---

> > ### Comment · Reviewer_9Q3m · 2023-09-12
> > **Response to Authors**
> >
> > ```We have experimented with scatter plots that show the trade-off between privacy (one for MSE and another for AUC on the x-axis) and utility (accuracy on the y-axis), in a way that graphically shows what we have described in words in Sec 6.2.```
> >
> > Could you please add such a plot to the paper? I am assuming that the plot would be added in the final version.

---

> > > ### Author Response · Authors · 2023-09-15
> > >
> > > Yes, we will add the plots in the final version.

---

### Decision · Action_Editors · 2023-10-02

**Recommendation:** Reject

**Comment:**

Given the concerns raised by the reviewers and the promised revisions, I feel the paper would benefit from a major revision.

This shall address the concerns raised by the reviewers with changes promised by the authors, including
1. Adding detailed proofs of the theorems.
2. Adding figures to present the results in a way that allows easy evaluation of the privacy-utility trade-off of the different methods.
3. Address the question on privacy analysis depending on disjoint batches. All DP-SGD work during past 5+ years uses composition theorems or privacy accountants to allow using the same examples multiple times. Pretending that none of this work exists or is not relevant or out of scope is unacceptable.

In addition to the above concerns, I would highly encourage the authors to address these additional concerns:

4. Clarify and systematise the notation. Do not use $\epsilon$ for both privacy parameter and parameter of the VMF mechanism. Rather, use something like $\kappa$ as the VMF noise scale parameter, like Weggenmann & Kerschbaum. Then you can prove that mechanism with parameter $\kappa$ is $\epsilon d$-private with $\epsilon = \kappa$ or similar.
5. Start your presentation of the results with utility results, including a state-of-the-art DP-SGD baseline. I would encourage looking at recent literature and making sure your results are at least approximately comparable, and maybe even including some previously published results as additional benchmarks. arXiv:2204.13650 seems to provide a good review of recent results for CIFAR-10 with references. None of the MIA results matter if the models do not provide competitive accuracy.

As a final note, I accept your point that your method provides in some sense stronger guarantee with $\delta = 0$. However, as you are explicitly calibrating the privacy guarantees using attack susceptibility, adding an $(\epsilon, \delta)$-DP method to the mix should be very natural. (And any method with Gaussian mechanism anyway has $\delta > 0$.) If adding $\delta$ makes a significant difference, presumably it should show up in the attack results.

Finally, I would like to remind the authors of TMLR's review criteria: the work should be of interest to someone; and you must provide sufficient evidence to support your claims. If providing additional evidence to support your current claims feels too hard or impossible, you are also welcome to revise the claims to match the evidence. My interpretation of this is that it is OK to present a negative result of a method that does not improve the state-of-the-art in every sense, as long as you formulate the claims appropriately, provide sufficient evidence, and provide an analysis that someone would still find interesting.

**Audience:**

The reviewers indicate the paper could be interesting to some individuals in TMLR's audience, but given the lack of a demonstration where DirDP-SGD would provide any concrete benefit over state-of-the-art DP-SGD, I am not equally convinced.

**Claims And Evidence:**

The claims made in the submission are not supported by sufficient evidence.

First, the authors basically suggest their method as a potentially higher utility alternative to DP-SGD, yes they seem to fail to provide a fair comparison with a state-of-the-art DP-SGD approach. The accuracy of the Gaussian noise DP-SGD is very low - for example CIFAR-10 with $\epsilon=1$ can reach 60% accuracy, with clearly higher for larger $\epsilon$. This suggests that the claimed "state-of-the-art" baselines are far from that.

Second, the authors fail to include sufficiently detailed proofs of their theorems. They are promising to do this in the final version, but I feel that all the changes promised for this final version are so big they require another review.

Third, the presentation of the empirical results is so unreadable that it cannot be described as clear evidence for the empirical claims.

**Resubmission Of Major Revision:**

The authors may consider submitting a major revision at a later time.